# A unique serum IgG glycosylation signature predicts development of Crohn's disease and is associated with pathogenic antibodies to mannose glycan

Inflammatory bowel disease (IBD) is characterized by chronic inflammation in the gut. There is growing evidence in Crohn's disease (CD) of the existence of a preclinical period characterized by immunological changes preceding symptom onset that starts years before diagnosis. Gaining insight into this preclinical phase will allow disease prediction and prevention. Analysis of preclinical serum samples, up to 6 years before IBD diagnosis (from the PREDICTS cohort), revealed the identification of a unique glycosylation signature on circulating antibodies (IgGs) characterized by lower galactosylation levels of the IgG fragment crystallizable (Fc) domain that remained stable until disease diagnosis. This specific IgG2 Fc glycan trait correlated with increased levels of antimicrobial antibodies, specifically anti-*Saccharomyces cerevisiae* (ASCA), pinpointing a glycome–ASCA hub detected in serum that predates by years the development of CD. Mechanistically, we demonstrated that this agalactosylated glycoform of ASCA IgG, detected in the preclinical phase, elicits a proinflammatory immune pathway through the activation and reprogramming of innate immune cells, such as dendritic cells and natural killer cells, via an FcγR-dependent mechanism, triggering NF-κB and CARD9 signaling and leading to inflammasome activation. This proinflammatory role of ASCA was demonstrated to be dependent on mannose glycan recognition and galactosylation levels in the IgG Fc domain. The pathogenic properties of (anti-mannose) ASCA IgG were validated in vivo. Adoptive transfer of antibodies to mannan (ASCA) to recipient wild-type mice resulted in increased susceptibility to intestinal inflammation that was recovered in recipient FcγR-deficient mice. Here we identify a glycosylation signature in circulating IgGs that precedes CD onset and pinpoint a specific glycome–ASCA pathway as a central player in the initiation of inflammation many years before CD diagnosis. This pathogenic glyco-hub may constitute a promising new serum biomarker for CD prediction and a potential target for disease prevention.

✉e-mail: salomep@i3s.up.pt

Inflammatory bowel disease (IBD) is a complex chronic gut inflammatory disorder, comprising Crohn's disease (CD) and ulcerative colitis (UC), which predominantly affect young people in their most productive and active period of life. Despite several advances in IBD research, the etiology of IBD remains unclear, and there is still no cure[1,2]. Individuals undergoing the most advanced medical therapies frequently fail to achieve remission, losing response over time or displaying intolerance to treatments[3]. Therefore, efforts need to be directed toward understanding the earlier events that occur before disease onset so that targeted interventions can be developed.

In IBD, and predominantly in CD, like in other immune-mediated diseases, there is mounting evidence of a preclinical phase that starts years before diagnosis and is characterized by immunological changes that precede symptoms and perhaps even mucosal injury[4]. Having a better knowledge of this preclinical period may increase the potential for improved understanding of disease pathogenesis and improved therapies as well as disease prediction and prevention.

Antibodies to *Saccharomyces cerevisiae* (ASCAs) have been widely associated with IBD and are the most accurate biomarker of CD[5]. ASCAs target mannose glycans, which are a common polysaccharide component of the fungal cell wall, such as in *Candida albicans*, but they are also found on other microorganisms, such as bacteria, viruses and host glycoconjugates[6]. In fact, the structural similarity between glycans on microorganisms and those of host surface glycoconjugates may create a glycan mimicry between microorganisms and host cells. This may lead to the generation of glycan-specific antibodies that cross-react with human glycoantigens, thereby contributing to the development of immune-mediated diseases[6]. Initial results from the Proteomic Evaluation and Discovery in an IBD Cohort of Tri-service Subjects (PREDICTS) cohort revealed that pathways involving protein glycosylation, the innate immune response and the complement cascade were the most predictive markers and were detected in the sera of individuals with CD before clinical onset[7]. This evidence set the groundwork for investigating whether and how changes in glycosylation may constitute a primary event that triggers chronic inflammation associated with IBD development.

Glycosylation is a major post-translational modification that consists of the addition of carbohydrate structures (glycans) to essentially all cells[8]. Numerous studies demonstrate that glycosylation is substantially altered in major diseases, such as cancer, and in immune-mediated diseases, including IBD and lupus[8–14]. We have previously demonstrated that alterations in cellular glycosylation can trigger proinflammatory responses associated with increased susceptibility to severe forms of intestinal inflammation and early-onset colitis[8,10]. Together, these results highlight the key role of glycans in gut immunity and specifically in IBD immunopathogenesis.

A growing body of evidence has highlighted the prominent role of glycans in the regulation of humoral immune responses. IgG antibodies are the predominant antibody class in the circulation and are key effectors of the humoral immune system as they trigger leukocyte activation and inflammation[15,16]. Human IgG isolated from serum is composed of multiple glycoforms owing to the addition of a diverse repertoire of glycan structures on the IgG crystallizable fragment (Fc) region, a conserved domain responsible for modulating IgG effector functions through interactions with Fcγ receptors (FcγRs) on immune cells[17]. The presence of post-translational modifications in IgGs has been suggested to shape their activity and function[18]. Over 30 different glycan variations have been detected on circulating IgGs in healthy individuals[19], which reflects a tremendous heterogeneity in the IgG Fc glycome. Individuals diagnosed with IBD were shown to display altered glycosylation of IgG Fc antibodies that were associated with disease severity compared to healthy individuals[20,21]. The role of this IgG glycome heterogeneity or switching among the population in defining risk for the transition from healthy tissue to intestinal inflammation and IBD development years before diagnosis remains completely unknown.

Taking advantage of the unique preclinical PREDICTS cohort[22], with multiple longitudinal samples many years preceding IBD diagnosis, we identified a serum glycome signature on circulating antibodies (IgGs) detectable up to 6 years before CD diagnosis. This IgG glycome signature was specific for CD and was not observed in preclinical samples of UC. We further demonstrated that the altered IgG Fc glycome signature (specifically the agalactosylated IgG2 H3N4F1 glycoform), detected in a preclinical phase, was positively correlated with the presence of ASCA, unlocking a unique glycome–ASCA hub that predicts CD development years before diagnosis. In addition, we demonstrated that the preclinical ASCA IgG glycoform was able to activate innate immune cells toward a proinflammatory phenotype via an FcγR-dependent mechanism, promoting NF-κB signaling as well as *CARD9* expression and inflammasome activation in dendritic cells (DCs). These pathogenic properties of ASCA were confirmed by adoptive IgG transfer to wild-type (WT) mice that subsequently developed an increased susceptibility to intestinal inflammation, which was hampered in recipient FcγR-deficient mice.

## Results

### Changes in IgG Fc glycosylation predate CD diagnosis

Serum samples from individuals with CD, individuals with UC and healthy control (HC) individuals from four different time points (approximately 6, 4 and 2 years before diagnosis and at the time of diagnosis; Supplementary Table 1) were retrieved from the US Department of Defense serum repository. The glycosylation profile of the Fc region of IgG was evaluated by nano-liquid chromatography–tandem mass spectrometry (nano-LC–ESI–MS; Supplementary Fig. 1a). Glycan structures were grouped according to their glycan traits (Supplementary Table 2). Specific glycoforms of IgG Fc, containing predominantly less complex *N*-glycan structures, were consistently detected in the circulation in all preclinical CD samples up to 6 years before diagnosis (Fig. 1a). Individuals that later developed CD displayed a distinct and unique IgG Fc glycoprofile, characterized by an overall Fc agalacto-sylation of serum IgG1, IgG2 and IgG4 (predominantly the H3N4F1

**Fig. 1 | Alterations in IgG Fc glycosylation precede CD development and are associated with complicated disease. a**, Total IgGs from serum samples from *n* = 251 individuals with CD and *n* = 250 HC individuals at different time points (preclinical phase: 1–2, 4–6 and 6 years before CD diagnosis and at CD diagnosis matched for HCs) were isolated and analyzed by mass spectrometry. The association between glycan traits measured at different time points and CD onset is shown. The size of the bubble corresponds to the *P* value from two-sided *t*-tests from a logistic regression model ($-\log_{10}$ scale) after adjusting for sex, race and age, whereas bubble color corresponds to odds ratio. Only associations significant at a 10% false discovery rate are reported. **b**, Association between different glycan traits and the development of CD complications (depicted as days to CD complication development) assessed by Kaplan–Meier analysis. *P* values from two-sided log-rank tests are reported. **c**, Spearman's correlations between serologic markers and glycan traits associated with CD onset. Correlations are reported considering only individuals with CD for different time points before diagnosis. **d**, Spearman's correlations between serologic markers and glycan traits associated with CD onset. Correlations are reported considering only HC individuals for different time points before diagnosis. **e**, Estimated coexpression networks based on individuals with CD capturing the association between glycan traits and serologic markers. **f**, Box plot of IgG2 H3N4F1 stratified by ASCA IgA (ASCAA) and ASCA IgG (ASCAG) positivity for individuals with CD for different time points before diagnosis. The number of individuals with CD in **f** for each time point is 200 at diagnosis (Dx), 116 at −1 to −2 years, 165 at −2 to −4 years and 201 at −6 years before diagnosis. In the box plot, the bottom and top hinges correspond to the first and third quartiles (the 25th and 75th percentiles), with median levels represented by a horizontal line. The top (bottom) whisker extends from the hinge to the largest (smallest) value no farther than 1.5× interquartile range from the hinge. Data beyond the end of the whiskers are defined as outliers and are plotted individually. *P* values from two-sided *t*-tests are reported.

glycoform; G0F), compared to HC individuals, which exhibited an overall increase in digalactosylation of serum IgG Fc (H5N4F1; G2F; Fig. 1a). These distinct glycoprofiles of IgG Fc were detected at the farthest time point to the diagnosis (Fig. 1a). We also demonstrated that a specific agalactosylated IgG2 Fc glycoform (namely H3N4F1), present in the sera of individuals with CD up to 6 years before diagnosis, significantly

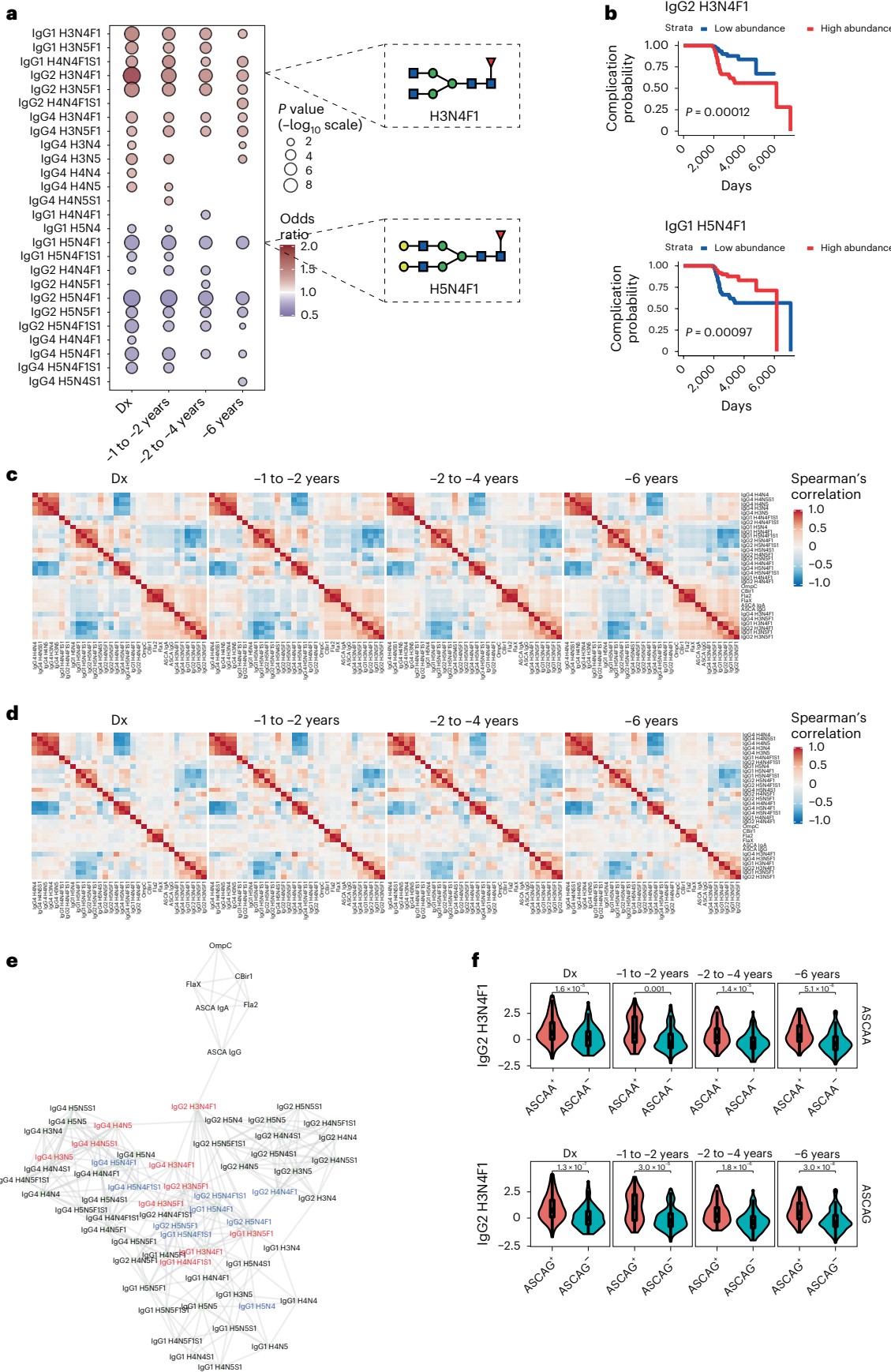

correlated with a complicated CD phenotype at diagnosis, in contrast to a digalactosylated IgG Fc glycoform (Fig. 1b and Supplementary Fig. 1b). This agalactosylated profile seems to be specific for serum IgG because no major alterations were found for agalactosylated forms of serum IgA at the preclinical stage (Supplementary Fig. 1c). By contrast, the IgG Fc glycoprofile in individuals with UC only changed at the time of diagnosis (Supplementary Fig. 2).

We then correlated the levels and profiles of IgG Fc glycans with serum antimicrobial antibodies that have previously been described in the preclinical phase of CD, namely ASCA, anti-flagellin (Fla2 and FlaX) and anti-*Escherichia coli* outer membrane porin (OmpC)[7]. We observed that the IgG Fc glycosignature that precedes CD development clustered with antimicrobial antibodies in a stable and consistent way and up to 6 years before diagnosis (Fig. 1c). This correlation between the altered serum IgG Fc glycome and antimicrobial antibodies was only observed in individuals with CD and not in HC individuals (Fig. 1d). In addition, using a network analysis of the association between IgG glycoprofiles and serum antimicrobial antibodies in individuals with CD at the farthest time from diagnosis (6 years before), we demonstrated a unique connection between the IgG2 H3N4F1 glycoform and ASCA IgGs, pinpointing a serum glycome–ASCA hub that predates clinical CD development (Fig. 1e). Indeed, IgG2 H3N4F1 levels were significantly higher among individuals positive for ASCA IgA and ASCA IgG for all time points before diagnosis (Fig. 1f), supporting its potential as a marker of subsequent CD development.

## Preclinical ASCA elicits proinflammatory immune responses

ASCA levels from a subset of samples from the PREDICTS cohort were compared to those from individuals with full-blown, established CD (Supplementary Table 3). ASCA levels gradually increased throughout the preclinical phase of disease until reaching established CD (Supplementary Fig. 1d). The same profile is found for ASCA titers, specifically analyzed for a subset of samples (Supplementary Fig. 1e).

To explore whether ASCA IgG Fc glycosylation may contribute to a proinflammatory response years before diagnosis, we tested the ability of ASCA IgG to activate immune cells such as DCs (Supplementary Fig. 3a) because DCs express FcγRIIa, which is known to interact with Fc glycans on IgG[23]. The ASCA IgG glycoform from individuals with CD before diagnosis induced a clear DC activation, as demonstrated by the upregulation of CD86 and *FCGR2A* expression (the latter of which encodes FcγRIIa). This innate immune cell activation induced by the ASCA IgG glycoform was found to occur up to 6 years before CD diagnosis (Fig. 2a,b). To further validate that the interaction between ASCA IgG Fc and DCs was mediated by FcγR, incubation with a FcγRIIa-blocking

antibody (anti-CD32a) was performed (Supplementary Fig. 3b), and a decrease in DC activation was observed (Fig. 2c). To exclude any potential effect of IgG aggregation in FcγR binding and activity, we measured IgG aggregation after isolation in the different (pre)clinical conditions. No major differences were observed for all the groups (Supplementary Fig. 3c,d).

This effect observed in innate immune cell activation was accompanied by a significant increase in the production of proinflammatory cytokines, such as interleukin-1β (IL-1β), IL-6, IL-8 and tumor necrosis factor (TNF), by DCs compared to IgGs from HC individuals (Fig. 2d–h). No differences were found in the production of the anti-inflammatory cytokine IL-10 (Fig. 2i). Moreover, FcγRIIa blockade resulted in a general decrease in cytokine production, particularly IL-6 and TNF, in preclinical individuals with CD (Supplementary Fig. 3e–i). Together, these results reveal the proinflammatory properties of ASCA IgG glycoforms in shaping innate immune cell activation through Fc–FcγRIIa interactions during the preclinical phase of CD. The proinflammatory properties of ASCAs were also observed in natural killer (NK) cells, with increased granzyme B secretion and CD107a surface expression (Supplementary Fig. 4a–c). A slight nonsignificant increase in interferon-γ (IFNγ) production was also observed (Supplementary Fig. 4d).

To gain further insights into the immunological remodeling of DCs, NK cells and B cell-derived plasma cells that is involved in antibody production that occurs in the preclinical phase, we analyzed the cell-type signatures from the report by Newman and colleagues[24] using somalogics data generated within our consortium[7]. This analysis showed that markers associated with activated DCs and NK cells were found to be increased at the farthest time points from diagnosis in individuals with CD compared to other immune populations (Supplementary Fig. 5a,b). We also found that the expression levels of markers for plasma cells were significantly increased in the 2- to 4-year timeframe before diagnosis, supporting the existence of a potential remodeling of B cell-derived plasma cells, within the context of a subclinical inflammation, which may contribute to autoantibody production associated with intestinal inflammation (Supplementary Fig. 5c). This was complemented by a significant increase in the frequency of B cells and plasmablasts observed in first-degree relatives of individuals with CD compared to individuals diagnosed with CD, concomitant with an increasing trend in the frequency of plasma cells (Supplementary Fig. 5d–f and Supplementary Table 4). No major alterations were found in IgG-producing plasma cells in these individuals (Supplementary Fig. 5g). Together, these results support the remodeling of immunological pathways related to immune cell activation occurring during the transition from a healthy intestine to intestinal inflammation.

**Fig. 2 | ASCAs generated years before CD diagnosis can trigger proinflammatory immune responses. a**, DCs incubated with ASCAs from preclinical individuals (Pred_6y, $n = 44$) and individuals diagnosed with CD (Pred_Dx, $n = 44$; established CD (full-blown), $n = 18$) displayed increased CD86 expression (data are normalized to HC individuals; HC, $n = 62$); MFI, mean fluorescence intensity; w/o, without. **b**, *FCGR2A* expression is also increased in DCs incubated with ASCAs from preclinical individuals (Pred_6y, $n = 26$) and individuals diagnosed with CD (Pred_Dx, $n = 26$; established CD, $n = 11$) compared to HC individuals ($n = 34$). *FCGR2A* expression is shown normalized to the expression of the housekeeping (HK) gene 18S. **c**, CD86 expression on DCs was decreased after blockade of FcγRIIa (data are normalized to that of HC individuals; control: HC $n = 17$, Pred_6y $n = 17$, Pred_Dx $n = 15$, established CD $n = 7$; with CD32a inhibition: HC $n = 16$, Pred_6y $n = 17$, Pred_Dx $n = 15$, established CD $n = 7$). **d–i**, ASCAs from individuals before and after diagnosis of CD promoted a distinct proinflammatory profile on DCs compared to ASCAs from HC individuals ($n = 6$–36 per group). Data in **d** corresponds to the fold change compared to HC. Data in **e** show HC ($n = 23$), Pred_6y ($n = 16$), Pred_Dx ($n = 11$) and established CD ($n = 6$). Data in **f** show HC ($n = 35$), Pred_6y ($n = 33$), Pred_Dx ($n = 29$) and established CD ($n = 10$). Data in **g** show HC ($n = 36$), Pred_6y ($n = 23$), Pred_Dx ($n = 23$) and established CD ($n = 10$). Data in **h** show HC ($n = 31$), Pred_6y ($n = 23$), Pred_Dx ($n = 19$) and established CD ($n = 9$). Data in **i** show HC ($n = 12$),

Pred_6y ($n = 20$), Pred_Dx ($n = 25$) and established CD ($n = 10$). **j**, ASCAs from individuals before and after diagnosis with CD show decreased galactosylation (by ECA reactivity) compared to ASCAs from HC individuals ($n = 3$ per group; independent replicates). **k**, Ablation of galactose residues on IgGs from HC individuals (HC β-gal; $n = 19$) leads to increased CD86 expression similar to that imposed by ASCAs from individuals with CD (normalized to HC; HC, $n = 19$; Pred_6y, $n = 20$; Pred_Dx, $n = 17$). **l**, CD86 expression on DCs is lower when in contact with di-GlcNAc-specific IgGs than ASCAs (normalized to HC mannan; mannan: HC $n = 17$, Pred_6y $n = 15$, Pred_Dx $n = 13$, established CD $n = 4$; di-GlcNAc: HC $n = 17$, Pred_6y $n = 17$, Pred_Dx $n = 12$, established CD $n = 5$). **m**, DCs cocultured with di-GlcNAc-specific IgGs also display a decrease in TNF production (mannan: HC $n = 8$, Pred_6y $n = 7$, Pred_Dx $n = 7$, established CD $n = 4$; di-GlcNAc: HC $n = 6$, Pred_6y $n = 9$, Pred_Dx $n = 8$, established CD $n = 5$). Data in **c**, **l** and **m** were analyzed comparing treatments within each group by Mann–Whitney $t$-test. Data in **j** and **k** were analyzed comparing each condition with the control (HC) by one-way analysis of variance with an uncorrected Fisher's least significant difference test. Data on the remaining graphs were analyzed comparing each condition with the control (HC) by Kruskal–Wallis test with an uncorrected Dunn's test. $P$ values are shown in the graphs. Each data point represents the data from a single individual (biological replicates).

Next, our observation that serum IgG2 Fc agalactosylation (H3N4F1) was the only IgG glycosignature that specifically correlated with ASCAs predicting CD development (Fig. 1e) raised the question about the particular impact of the agalactosylated form of ASCAs in inducing early immune activation. To address this, we modulated the galactose levels on IgGs from HC individuals and evaluated the specific impact on DC activation. Levels of ASCA IgG galactosylation were confirmed by *Erythrina cristagalli* lectin (ECA) reactivity, which specifically recognizes galactose glycan residues (Fig. 2j). Total IgGs (including ASCAs) isolated from HC individuals ('healthy' galactosylated Fc profile)

were digested with β-galactosidase (β-gal) to mimic an agalactosylated ('pathogenic') IgG Fc glycosignature (Supplementary Fig. 4e). Degalactosylated ASCA IgG Fc from HC individuals led to a significant increase in DC activation, with CD86 expression reaching similar levels as those observed using ASCA IgG from individuals with CD (Fig. 2k). A similar activation profile was also observed for NK cells, in which removal of galactose from ASCA IgGs from HC individuals resulted in higher granzyme B secretion and LAMP-1 expression (Supplementary Fig. 4b,c).

Together, these findings suggest that reducing levels of galactosylation on ASCA IgG Fc switches its function toward a proinflammatory

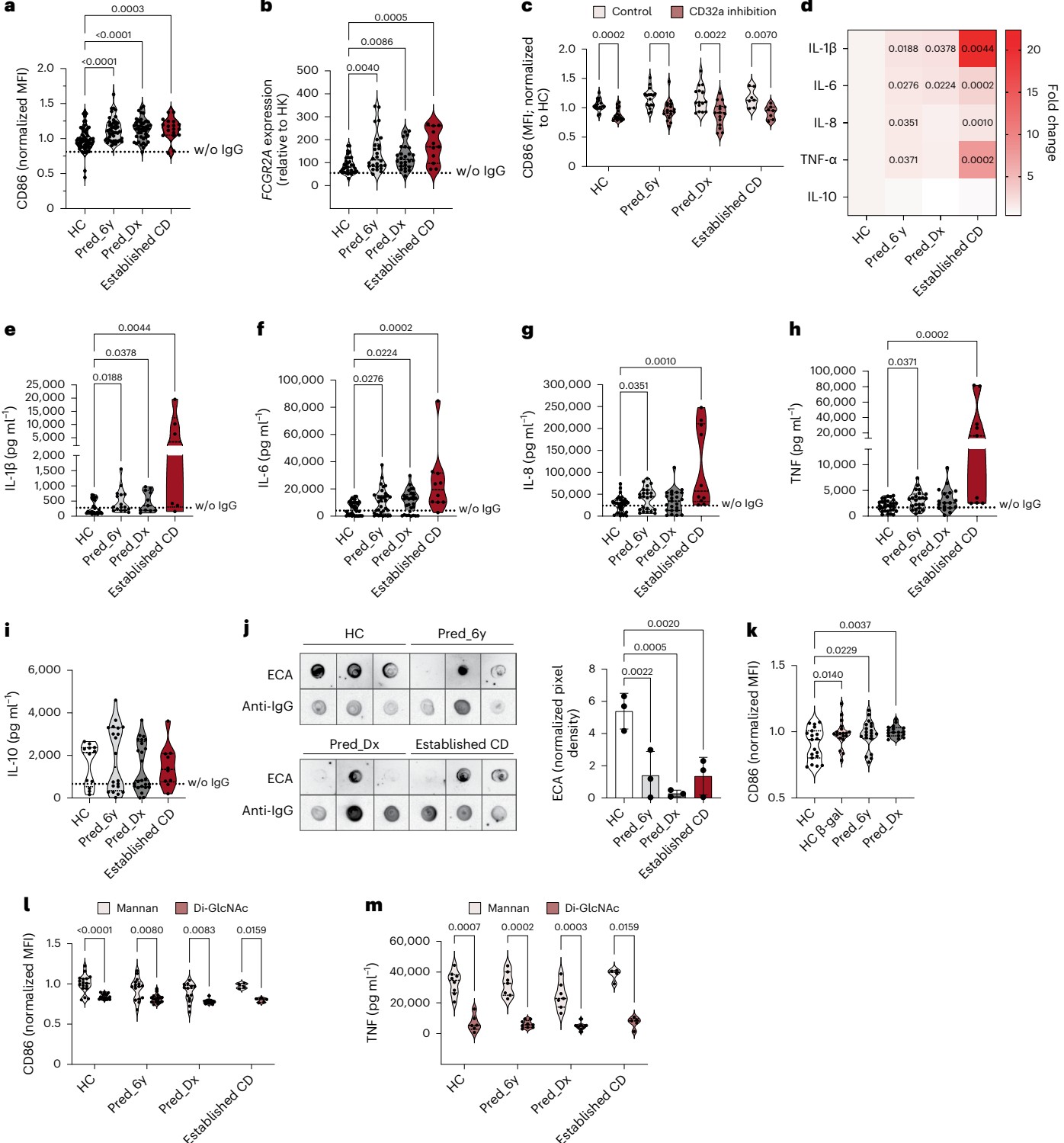

phenotype by triggering the activation of the innate immune response mediated by DCs and NK cells. This highlights a key role of ASCA IgG glycosylation as an important inducer of proinflammatory responses years before CD diagnosis.

We then explored whether the proinflammatory response was dependent not only on ASCA Fc galactosylation but also its specificity for mannose glycan–antigen recognition and not for other glycan epitopes. Mannan-unbounded IgGs from preclinical and diagnosed CD had no impact on DC activation compared to those from HC individuals, demonstrating the specific impact of mannan-specific ASCAs in activating DCs (Supplementary Fig. 6a). Furthermore, we observed that DC activation was significantly decreased in cocultures with anti-di-GlcNAc (a nonmannose glycan) compared to ASCA IgGs (Fig. 2l and Supplementary Fig. 6b,c), as was the secretion of TNF and IL-8 (Fig. 2m and Supplementary Fig. 6d). Incubation with other microbial patterns, in particular the yeast/bacteria-derived β-glucan, demonstrated that antibodies to β-glucan caused no differential activation of DCs (Supplementary Fig. 6e). The selective activation imposed by ASCA IgGs was also observed for NK cells (Supplementary Fig. 4f,g).

Overall, these results support the proinflammatory function of the ASCA–glycome hub detected years before disease diagnosis by demonstrating that mannan recognition through the IgG Fc agalactosylated profile triggers innate immune cell activation, initiating a proinflammatory process years before CD diagnosis.

### Preclinical ASCAs reprogram glycan-binding proteins in DCs

To assess, using an unbiased approach, the impact of ASCAs on the reprogramming of immunological pathways during preclinical CD, RNA-sequencing analysis of DCs cocultured with preclinical ASCA IgG glycoforms from different settings (HC, 6 years before diagnosis and at diagnosis) showed clear differential gene expression profiles compared to DCs incubated with ASCA IgGs from HC individuals and individuals with CD at the time of diagnosis (Fig. 3a). We found a significant increase in the expression of genes involved in proinflammatory signaling pathways, such as TNF, NF-κB and C-type lectin receptor pathways (Fig. 3b), up to 6 years before diagnosis compared to DCs incubated with ASCAs from HC individuals. Indeed, *TNF* was found to be one of the top genes significantly upregulated by ASCA IgGs in the preclinical phase (Fig. 3c), supporting the premise that ASCA IgGs act as an early trigger for the transition between a healthy intestine and intestinal inflammation. Additionally, one of the pathways shown to be upregulated by preclinical ASCA IgGs is the complement cascade (Fig. 3b), in particular *C1QA* and *C1QB* (Fig. 3c), which matches our results demonstrating that ASCA IgGs from individuals with CD exhibit an increase in C1q binding (Supplementary Fig. 3j) and agrees with the proinflammatory environment imposed by ASCA IgGs in pre-CD. This complement activation is apparently independent of galactose levels on ASCA IgGs (Supplementary Fig. 3k).

Taking into consideration the data provided by the RNA sequencing of DCs, we analyzed the impact of ASCA glycoforms on the modulation of DC activation programs. DCs are equipped with glycan-binding proteins (C-type lectins), such as DC-SIGN and dectin-2, which enable them to recognize glycan structures such as mannose glycans, thus shaping their immune response[25–27].

Agalactosylated ASCA IgGs from individuals before CD diagnosis imposed a significant overexpression of DC-SIGN on DCs at the cell surface (Fig. 3d). Dectin-2 also showed a significant overexpression on DCs at the time of diagnosis (Fig. 3e). The increased expression of these glycan-binding proteins on DCs after interaction with ASCA IgG glycoforms appears to occur very early in disease development and is maintained until CD onset, likely shaping innate immune function.

The NF-κB signaling pathway (a major downstream player of DC-SIGN and dectin-2 and FcγR activation on DCs) was also identified as significantly upregulated in the signaling pathway analysis (Fig. 3b). We also showed increased phospho-NF-κB activation in DCs cocultured with ASCA IgGs, predominantly up to 6 years before diagnosis (Fig. 3f), suggesting that the NF-κB signaling pathway is modulated by ASCA IgG glycoprofiles in the preclinical phase of CD. This is in accordance with the increased production of several cytokines known to be produced after NF-κB activation (Fig. 2d–h). For individuals with CD with established disease, no major alterations were found (Fig. 3f), which can be associated with the fact that these individuals are not naive to therapy, and thus the immunomodulatory treatment may be influencing their immune behavior.

Importantly, FcγRIIa on DCs can activate the NF-κB pathway and its intermediate partners, such as the NLRP3 inflammasome[26]. Indeed, *NLRP3* expression in DCs was significantly increased after interaction with ASCA IgGs from individuals with CD up to 6 years before diagnosis and was maintained until the establishment of CD (Fig. 3g). Accordingly, the increased production of IL-1β by DCs (Fig. 2e) also corroborates the activation of the *NLRP3* inflammasome. FcγRIIa and several C-type lectins expressed on DCs can engage CARD9 for NF-κB activation and proinflammatory gene transcription[26,28–30]. In fact, genetic mutations in *CARD9* are described in CD, in which predisposing variants are associated with increased expression of *CARD9* mRNA and

---

**Fig. 3 | ASCA Fc glycoforms from individuals before CD diagnosis shape the expression of proinflammatory signaling pathways and glycan-binding proteins in DCs, leading to *NLRP3* and *CARD9* expression. a**, Heat map of the relative expression values (z score of each gene across samples) for the top six upregulated KEGG pathways found in DCs cocultured with ASCA IgGs from individuals at the preclinical phase (Pred_6y, $n = 4$) versus ASCA IgGs from HC individuals ($n = 5$). In addition to Pred_6y and HC individuals, the heat map also shows the expression of these genes in DCs cultured with ASCA IgGs from individuals at diagnosis (Pred_Dx, $n = 5$). Only differentially expressed genes are shown, excluding genes belonging to more than one pathway. **b**, Dot plot of the overrepresentation analysis (KEGG ontology) for Pred_6y versus HC. The plot represents the top six enriched pathways (false discovery rate < 0.05), excluding human disease pathways. The diameter of the dot indicates the number of upregulated genes (false discovery rate < 0.05) belonging to the pathway. The color code indicates the adjusted *P* value for the pathway. Overrepresentation uses a hypergeometric test corrected for multiple testing using the Benjamini–Hochberg method to determine the statistical significance of the upregulated differentially expressed genes in each Gene Ontology term. **c**, Volcano plot depicting differentially expressed genes in DCs cocultured with ASCAs from Pred_6y ($n = 4$) versus HC ($n = 5$) individuals. Red dots represent genes expressed at higher levels in Pred_6y individuals. Black dots represent genes below the cutoff of significance ($|\log_2$ (fold change)$| \geq 0.5$ and adjusted

*P* values of ≤0.05). DESeq2 uses negative binomial generalized linear models for the differential analysis of count data and the Wald test with multiple correction (Benjamini–Hochberg method) for significance testing. The $\log_2$ (fold change) values were shrunken with the apeglm method to increase the signal-over-noise ratio of the effect size. **d**, ASCAs from preclinical (Pred_6y, $n = 30$) and at-diagnosis CD samples (Pred_Dx, $n = 30$; established CD, $n = 16$) can induce the surface expression of DC-SIGN on DCs (HC, $n = 36$). **e**, DCs display increased expression of dectin-2 after incubation with ASCAs from individuals with CD (Pred_Dx, $n = 35$; established CD, $n = 15$; HC, $n = 43$); no major alterations were found for Pred_6y ($n = 31$). **f**, DCs cultured with ASCAs from preclinical samples show increased expression of phospho-NF-κB (HC, $n = 8$; Pred_6y, $n = 8$; Pred_Dx, $n = 8$; established CD, $n = 5$; independent replicates). **g**, ASCAs from preclinical individuals and individuals with established CD promoted the upregulation of *NLRP3* expression in DCs (HC, $n = 29$; Pred_6y, $n = 26$; Pred_Dx, $n = 30$; established CD, $n = 11$). **h**, Similar profiles were found for *CARD9* expression (HC, $n = 29$; Pred_6y, $n = 23$; Pred_Dx, $n = 29$; established CD, $n = 9$). Data are presented as mean ± s.d. and were analyzed comparing each condition to the control (HC). Data presented in **f** were analyzed by paired one-way ANOVA with uncorrected Fisher's least significant difference test compared to HC and are shown as mean ± s.d.; data on the remaining graphs were analyzed by a Kruskal–Wallis test with an uncorrected Dunn's test. *P* values are shown in the graphs. Each data point represents the data from a single individual (biological replicates).

---

activation of the immune response[31,32]. We showed that *CARD9* expression was significantly increased in DCs cocultured with ASCA IgGs (Fig. 3h). This increase in expression was more robust after stimulation with preclinical ASCAs (Fig. 3h), suggesting the relevance of ASCA IgG glycans in early DC activation and in triggering of inflammatory processes, whereas other pathways may have greater contributions in fully established CD.

## Anti-mannan IgGs increase susceptibility to mouse colitis

To further validate the pathogenic properties of ASCA IgGs in inflammation initiation and disease development, we performed a proof-of-concept experiment to evaluate the ability of anti-mannan to trigger IBD in vivo. WT mice were immunized with mannan to stimulate the production of anti-mannan (ASCA-like) IgGs (Fig. 4a). Mannan immunization resulted in significantly increased levels of serum ASCA

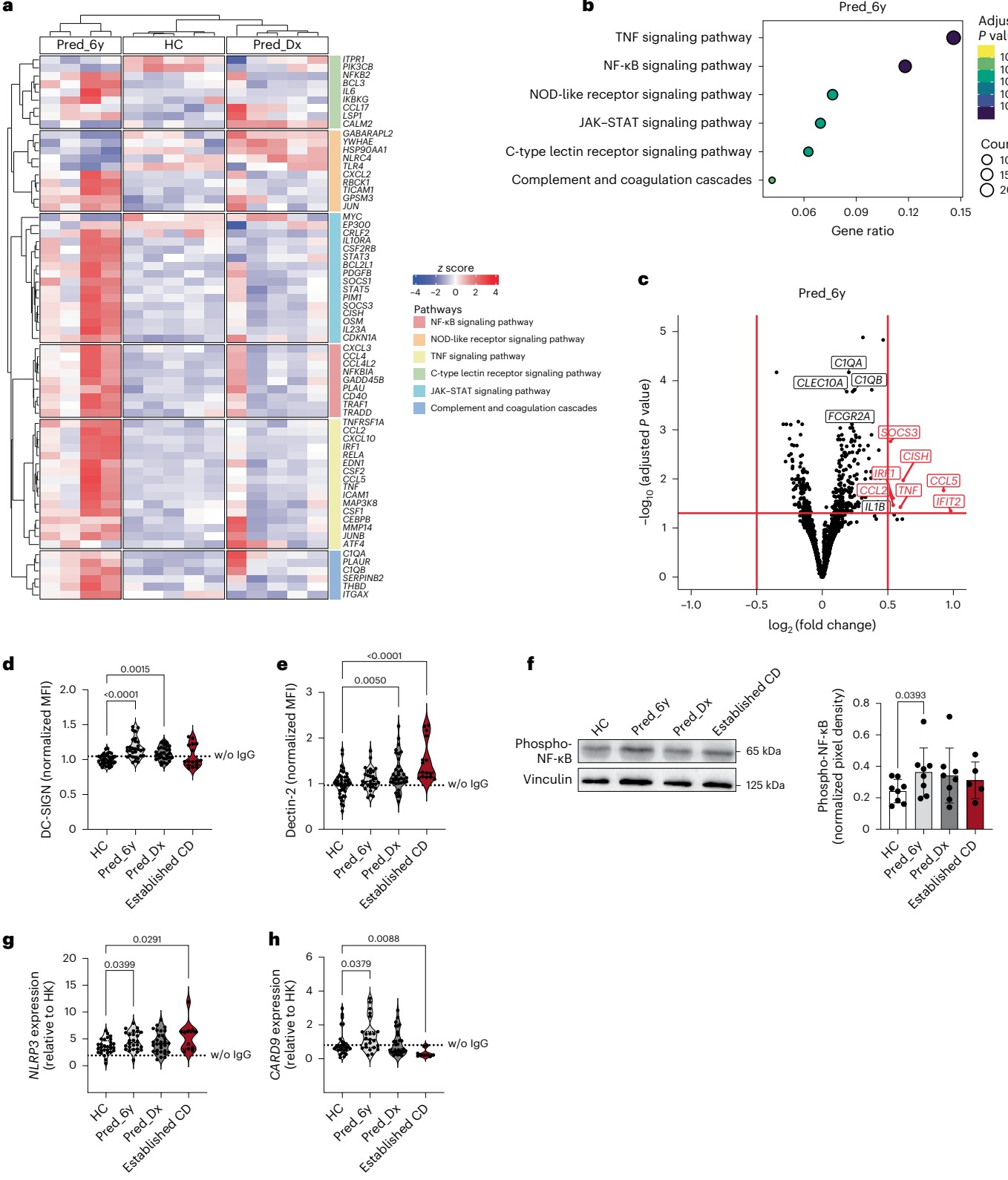

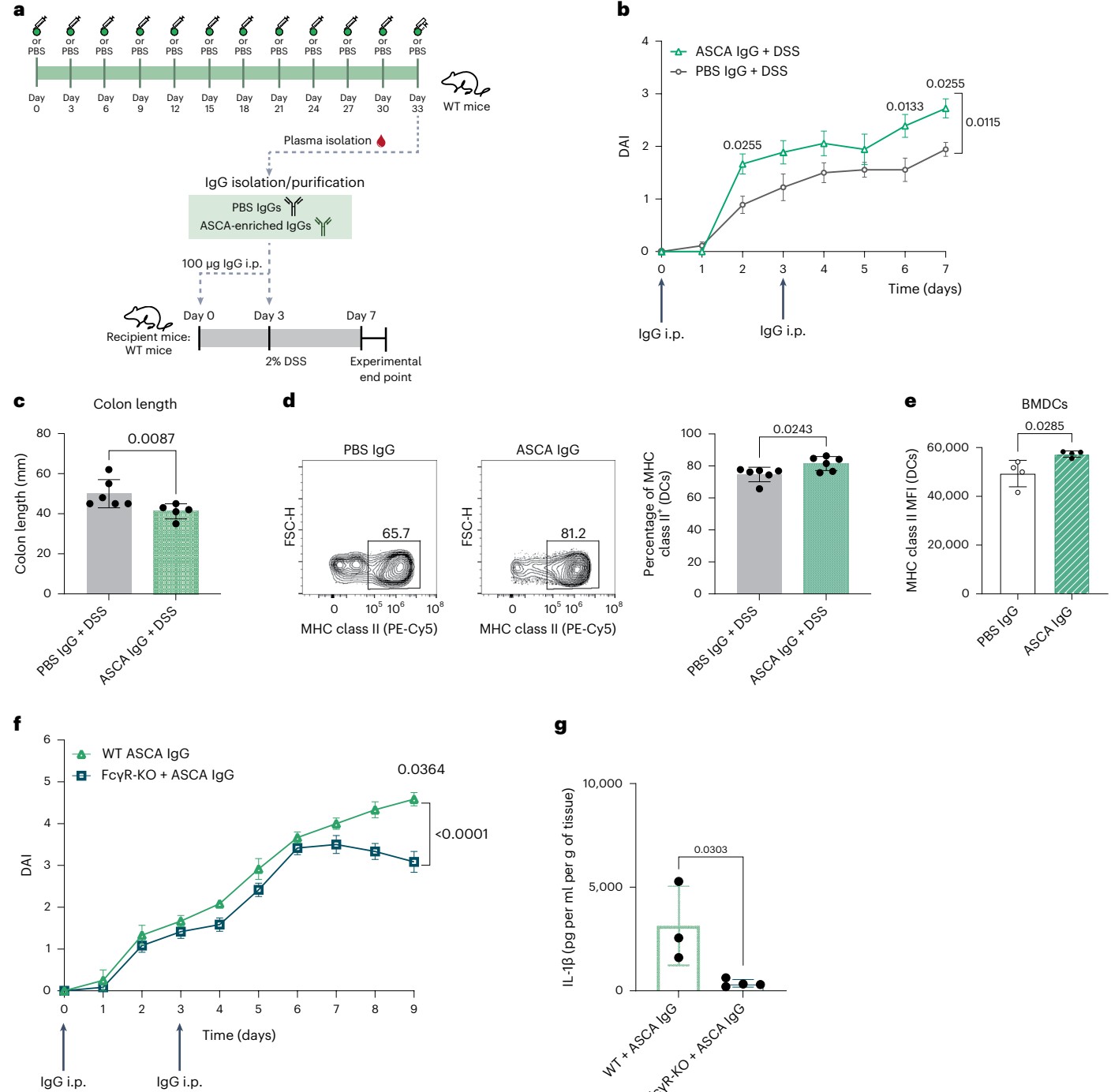

**Fig. 4 | Antibodies to mannan impose an increased susceptibility to colitis in mice. a**, WT mice were inoculated subcutaneously with mannan to promote the generation of antibodies to mannan (ASCA-like). Control mice were injected with PBS. Total IgGs were collected from both groups, and recipient mice were inoculated twice with 100 µg of total IgGs (ASCA enriched or from PBS-injected mice (control)), while colitis was chemically induced by administration of 2% DSS in drinking water; i.p., intraperitoneal. **b**, Mice inoculated with ASCA IgGs (*n* = 6) display increased susceptibility to colitis by showing increased DAI compared to PBS IgG-inoculated mice (*n* = 6). Data are presented as mean ± s.e.m. **c**, ASCA IgG-inoculated mice (*n* = 6) also display shortening of the colon compared to control animals (*n* = 5). **d**, ASCA-inoculated mice (*n* = 6) display increased frequencies of MHC class II⁺ cells in the colonic tissue after the induction of colitis compared to control animals (*n* = 6). **e**, BMDCs from WT mice were cocultured with

mannan-specific IgGs (mouse ASCA IgGs; *n* = 4) or nonspecific IgGs (mouse PBS IgGs; *n* = 4), and MHC class II expression (activation) was assessed by fluorescence-activated cell sorting (FACS). **f**, FcγR-KO mice inoculated with ASCA IgGs (*n* = 4) showed decreased susceptibility to colitis compared to WT mice (*n* = 4). Data are presented as mean ± s.e.m. **g**, FcγR-KO mice (*n* = 4) displayed decreased levels of IL-1β in the supernatants of colonic explants compared to WT mice (*n* = 3). Scatter dot plots are presented as mean ± s.d. Data presented in **b** and **f** were analyzed by two-way ANOVA with a Šídák's post-test, data presented in **c** were analyzed by two-tailed Mann–Whitney test, and data in the remaining graphs were analyzed by two-tailed unpaired *t*-test. *P* values are shown in the graphs. In **e**, each data point represents a technical replicate. For the remaining figures, each data point represents the data from a single individual (biological replicates).

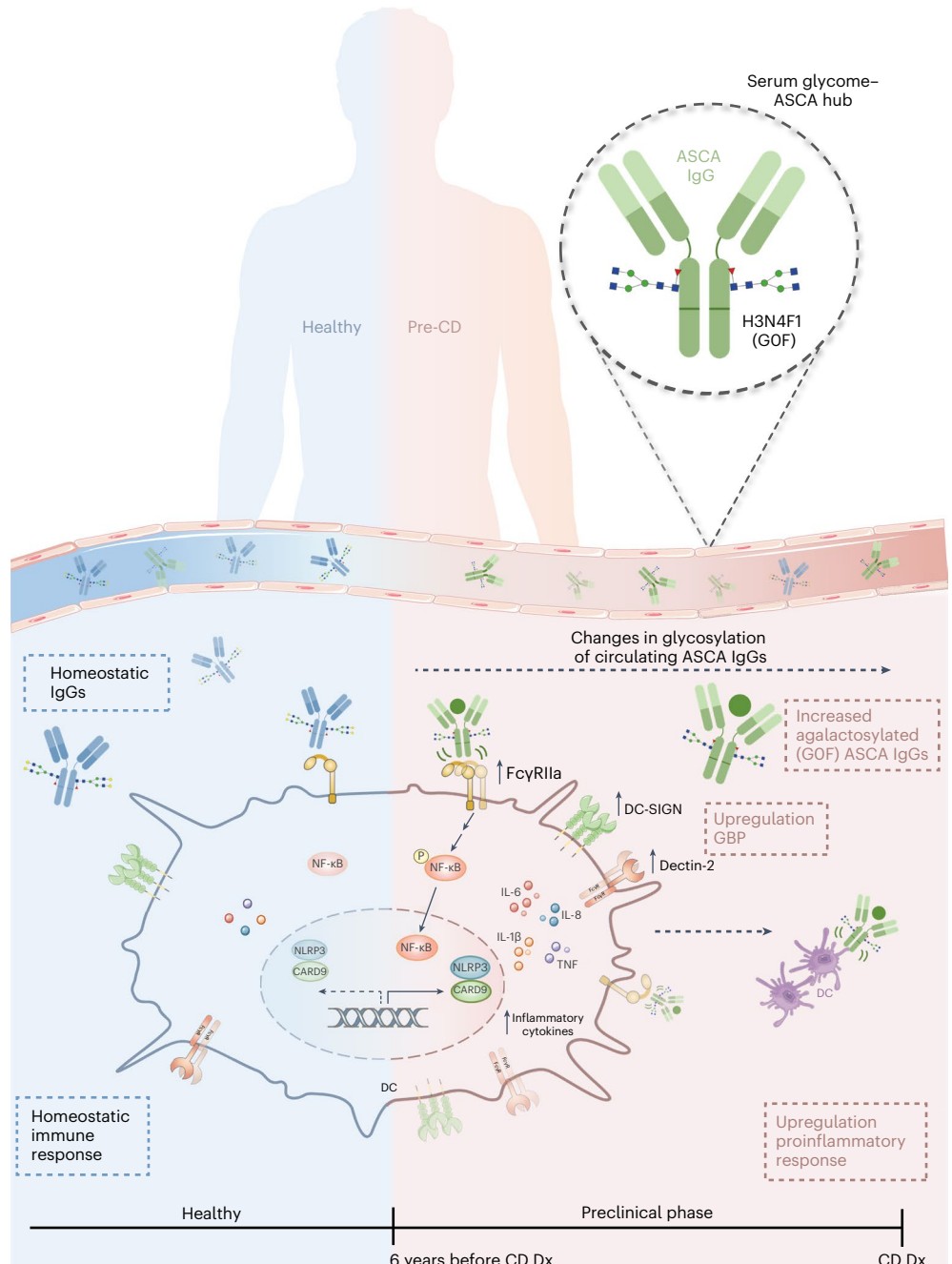

**Fig. 5 | ASCA agalactosylation precedes CD development, contributing to the initial steps of intestinal inflammation.** Alterations in IgG Fc glycosylation were found up to 6 years before CD diagnosis. This altered glycosignature (IgG2 H3N4F1; G0F) was identified as significantly associated with antimicrobial antibodies, in particular ASCAs. These antibodies to mannan, even those from years before diagnosis, also contribute to intestinal inflammation by shaping the profile of immune cells toward proinflammation. ASCAs were also demonstrated to modulate the expression of glycan-binding proteins (GBP) on the surface of DCs. Moreover, these antimicrobial antibodies found 6 years before diagnosis induced phosphorylation of NF-κB, as well as the expression of *NLRP3* and *CARD9*. This work identified a new biomarker for CD prediction, as this altered glycoform of IgG Fc is detected in the circulation many years before disease diagnosis. Additionally, we detailed an axis mediated by ASCA IgG glycoforms that mechanistically is able to trigger an inflammatory event.

IgGs (Supplementary Fig. 7a). No significant alterations in intestinal permeability after mannan immunization were observed (Supplementary Fig. 7b). Serum was collected from immunized mice, and ASCA-enriched IgGs were isolated (along with PBS-immunized IgGs from the control group). Analysis of the serum IgG glycome showed that no major differences were found in terms of IgG glycan composition between immunized and nonimmunized mice (Supplementary Fig. 7c). Moreover, no differences were observed in circulating ASCA IgGs after dextran sulfate sodium (DSS) treatment for colitis induction

(Supplementary Fig. 7d). This was expected because WT mice, with no basal genetic alterations in terms of glycosylation machinery on B cells, were used for mannan immunization and ASCA production. ASCA-enriched IgGs were then transferred into recipient C57BL/6 WT mice, followed by DSS-induced colitis (Fig. 4a).

Mice inoculated with ASCA IgGs displayed higher susceptibility to and severity of colitis than mice in the control group (Fig. 4b). Indeed, ASCA-treated mice developed an early disease onset and exhibited a more severe phenotype with a higher disease activity index (DAI;

Fig. 4b and Supplementary Table 5) and a more marked shortening of the colon (Fig. 4c). Analysis of lamina propria leukocytes demonstrated an increase in the expression of major histocompatibility complex class II⁺ (MHC class II⁺) DCs (Fig. 4d), suggesting the impact of ASCA IgGs on the activation profile of DCs in vivo. Additionally, coculture of ex vivo ASCA IgGs from mannan-immunized mice with bone marrow-derived DCs (BMDCs) from WT mice showed a similar increase in the expression of MHC class II (Fig. 4e). A similar trend in activation imposed by ASCA IgGs was found in NK cells, with increased expression of granzyme B (Supplementary Fig. 7e). The frequency of IFNγ-producing NK cells was also increased; however, this finding did not reach statistical significance (Supplementary Fig. 7f).

To explore the biological effect of ASCA IgGs on imposing an inflammatory environment in an Fc–FcγR-dependent manner, we used a mouse model deficient for FcRγ (*Fcgr1/Fcgr2/Fcgr3/Fcgr4*-KO; hereafter referred to as FcγR-KO). WT and FcγR-KO mice were inoculated with ASCA IgGs isolated from mannan-immunized mice, and colitis was induced. FcγR-KO mice inoculated with ASCA displayed a significantly reduced susceptibility to colitis, with decreased DAI values compared to WT mice (Fig. 4f and Supplementary Fig. 7g) and lower levels of IL-1β in the supernatants of colonic explants (Fig. 4g). TNF levels were also decreased, despite not reaching statistical significance (Supplementary Fig. 7h). Overall, we demonstrated the relevance of ASCA IgG Fc–FcγR interactions in imposing a proinflammatory environment associated with intestinal inflammation, supporting the pathogenic effects of ASCA IgGs in vivo.

We further assessed whether oral supplementation with mannan would promote similar susceptibility to colitis. Oral immunization with mannan was not effective in inducing increased serum ASCA IgG levels compared to nontreated mice (Supplementary Fig. 8a,b), and, expectedly, no major alterations were found in colitis susceptibility (Supplementary Fig. 8c) nor colon length (Supplementary Fig. 8d), demonstrating that increased levels of ASCA IgGs are intricately related to increased susceptibility to colitis. Overall, our data suggest that ASCA IgGs may not only act as a classical biomarker for IBD but also play a key role in the inceptive stages of initiation of a proinflammatory response in the gut, potentially triggering a more aggressive inflammatory response.

## Discussion

Recent evidence obtained from the PREDICTS cohort revealed that changes in glycosylation pathways can be detected in serum years before CD diagnosis, with protein glycosylation being among the top three markers/pathways that appeared to be activated in the preclinical phase[7]. Moreover, we previously demonstrated that changes in the expression of complex branched *N*-glycans at the surface of the gut mucosa regulate the immune response associated with IBD immunopathogenesis[8,10,12,33]. We also recently described that neutralizing antibodies to granulocyte–macrophage colony-stimulating factor (GM-CSF) were found to be specific for a glycosylated form of GM-CSF, which is detectable years before CD diagnosis[34].

A seminal observation in the 1990s described that glycan structures from *S. cerevisiae*, specifically mannosylated glycans, elicit an antibody response associated with CD development[35]. Elevated levels of antibodies to yeast glycans (that is, ASCAs) are still the most sensitive and specific serologic marker for CD and are detectable years before diagnosis[36]. Accordingly, findings from the Crohn's and Colitis Canada Genetics Environment Microbial Project cohort (a study on first-degree relatives of individuals with CD) showed that ASCAs are highly predictive of CD development and can be detected years before diagnosis[37]. In addition, data from the PREDICTS cohort also showed that ASCA levels predict complicated disease at diagnosis[38].

Here, we disclose the proinflammatory properties and disease-initiating capacity of an altered serum IgG glycome years before diagnosis. We identified a specific glycome–ASCA hub that appears to act as an early event that triggers innate immune response activation, reprogramming immunological pathways toward inflammation many years before clinical CD diagnosis (Fig. 5). Importantly, this glycome–ASCA pathway was validated in vivo, in which the adoptive transfer of ASCA IgGs led to early-onset disease and a remarkable increased susceptibility to colitis.

We also pinpointed a precise agalactosylated IgG signature in CD (IgG2 H3N4F1) as one of the top IgG Fc glycan traits that uniquely connects with increased ASCA levels up to 6 years before diagnosis. Although our understanding of the regulation of protein glycosylation is limited, recent studies demonstrated that gene networks that regulate glycosylation of different proteins can be very different[39]. A large study on over 12,000 individuals identified 27 genetic loci that associate with IgG glycosylation, including some that are known risk factors for IBD[40]. Therefore, specific glycovariants on IgG that associate with IBD development are revealed in this study.

We demonstrated that ASCA IgG is more than a bystander biomarker of IBD and plays an effective immunomodulatory function in the initial steps of inflammation years before CD diagnosis. We showed that ASCA IgG glycoforms interact with DCs and NK cells through Fc–FcγR interactions, instructing a proinflammatory response. Mechanistically, this interaction was demonstrated to reprogram DCs toward a proinflammatory phenotype, promoting the expression of genes associated with several proinflammatory signaling pathways such as TNF, NF-κB and C-type lectin receptor signaling pathways. Concomitantly, it also led to increased expression of glycan-binding proteins such as DC-SIGN, together with increased NF-κB signaling and *NLRP3* and *CARD9* expression and increased production of IL-1β and IL-8. Remarkably, this proinflammatory response imposed by ASCA IgG glycoforms is mannose glycan specific, dependent on galactosylation levels of ASCA IgG Fc and, even more importantly, occurs in a preclinical phase (Fig. 5). The in vivo pathogenic effect of ASCA-like IgGs was also demonstrated. This susceptibility is considerably ameliorated in FcγR-KO mice, supporting the biological relevance of ASCA Fc–FcγR interactions in triggering the inflammatory process. Overall, our results highlight the pathogenic properties of ASCAs in preclinical CD.

Agalactosylated IgGs have been associated with many immune-mediated and infectious diseases, such as coronavirus disease 2019 (refs. 15,16,41–43), which is consistent with our results. In addition, the effect of ASCA Fc–FcγRIIa interactions associated with DC activation and modulation of glycan-binding proteins may represent a synergistic mechanism imposed by ASCA IgG Fc glycoforms toward equipping DCs with proinflammatory functions, an issue that should be further explored.

Together, here, we identified an immunological program underlined by an ASCA–glycome hub that occurs years before CD diagnosis and can trigger an inflammatory response associated with the transition from healthy tissue to intestinal inflammation. This study not only highlights a new preclinical biomarker of CD development associated with CD pathogenesis but also helps define a new target for future preventive strategies for IBD and other immune-mediated diseases.

## Online content

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

Joana Gaifem[1,17], Cláudia S. Rodrigues[1,2,17], Francesca Petralia [3], Inês Alves[1], Eduarda Leite-Gomes[1,2], Bruno Cavadas[1], Ana M. Dias[1], Catarina Moreira-Barbosa[4], Joana Revés[5], Renee M. Laird[6,7], Mislav Novokmet[8], Jerko Štambuk[8], Siniša Habazin [8], Berk Turhan [3], Zeynep H. Gümüş [3,9], Ryan Ungaro[10], Joana Torres [5,11,12], Gordan Lauc[8,13], Jean-Frederic Colombel [14], Chad K. Porter[15] & Salomé S. Pinho [1,2,16] ✉

[1]i3S, Institute for Research and Innovation in Health, University of Porto, Porto, Portugal. [2]ICBAS, School of Medicine and Biomedical Sciences, University of Porto, Porto, Portugal. [3]Department of Genetics and Genomic Sciences, Icahn Institute for Data Science and Genomic Technology, Icahn School of Medicine at Mount Sinai, New York, NY, USA. [4]Hospital da Luz Learning Health, Luz Saúde, Lisbon, Portugal. [5]Division of Gastroenterology, Hospital Beatriz Ângelo, Loures, Portugal. [6]Operationally Relevant Infections Department, Naval Medical Research Command, Silver Spring, MD, USA. [7]Henry M. Jackson Foundation for Military Medicine, Inc., Bethesda, MD, USA. [8]Genos Glycoscience Research Laboratory, Zagreb, Croatia. [9]Precision Immunology Institute, Icahn School of Medicine at Mount Sinai, New York, NY, USA. [10]Henry D. Janowitz Division of Gastroenterology, Department of Medicine, Icahn School of Medicine at Mount Sinai, New York, NY, USA. [11]Faculty of Medicine, University of Lisbon, Lisbon, Portugal. [12]Division of Gastroenterology, Hospital da Luz, Lisbon, Portugal. [13]University of Zagreb Faculty of Pharmacy and Biochemistry, Ante Kovačića, Zagreb, Croatia. [14]Department of Medicine, Division of Gastroenterology, Icahn School of Medicine at Mount Sinai, New York, NY, USA. [15]Translational and Clinical Research Department, Naval Medical Research Command, Silver Spring, MD, USA. [16]Faculty of Medicine, University of Porto, Porto, Portugal. [17]These authors contributed equally: Joana Gaifem, Cláudia S. Rodrigues. ✉e-mail: salomep@i3s.up.pt

## Methods

### Cohort description and selection criteria

This study was approved as nonhuman subjects research (PJT 19-08) by the Naval Medical Research Command. Serum samples from 251 CD, 249 UC and 250 HC cases were retrieved from the US Department of Defense Serum Repository. From almost all individuals (741 of 750), four serum samples were retrieved comprising four specific time points, one corresponding to the nearest sample available closest to diagnosis (approximately 1 year; identified here as Dx) and three more samples collected before the diagnosis of IBD and separated by approximately 2-year intervals. Complicated disease was categorized as previously described[38]. Samples from HC individuals with no medical record of IBD, rheumatoid arthritis, celiac disease or colon cancer were selected to match the individuals with IBD and were collected as controls (Supplementary Table 1)[22]. As a positive control for established disease, 33 serum samples from individuals with established CD were obtained at Centro Hospitalar Universitário Santo António, Porto (Supplementary Table 3). For the analysis of plasma cells, peripheral blood mononuclear cells (PBMCs) from individuals with CD (collected at the time of diagnosis (inaugural disease)), first-degree relatives of individuals with CD and HC individuals were obtained both at Centro Hospitalar Universitário Santo António, Porto, Hospital Beatriz Ângelo, Loures, and Hospital da Luz, Lisbon (Supplementary Table 4). Ethical approval was obtained from the Ethical Committees of all hospitals.

### Isolation of IgG from human and mouse sera and ASCA quantification in human samples

Human and mouse serum samples were thawed and centrifuged at 1,620$g$ for 10 min. Samples were then diluted in PBS and filtered through 0.45-μm Supor AcroPrep 96-well filter plates. Serum samples were applied to the protein G plate and washed to remove unbound proteins. IgG was eluted from protein G using 0.1 M formic acid (pH 2.5) into a 96-deep-well plate and immediately neutralized with 1 M ammonium bicarbonate.

ASCA levels and titers (serial dilutions) were measured in human serum samples using a QUANTA Lite ASCA IgG enzyme-linked immunosorbent assay (ELISA; QUANTA), according to manufacturer's instructions.

### Glycopeptide preparation and nano-LC–ESI–MS

Human and mouse IgGs were treated with TPCK-treated trypsin (Promega) and purified using solid-phase extraction on C18 beads (Chromabond, Macherey-Nagel). IgG glycopeptides were eluted into a 96-well PCR plate with 200 μl of 20% acetonitrile by centrifugation at 105$g$ for 5 min. Eluates were dried by vacuum centrifugation and stored at −20 °C until analysis by MS. For human IgA, plasma samples were transferred into corresponding wells of Orochem filter plates containing 40 μl of IgA beads slurry (Thermo Scientific, IgA Affinity Matrix). Following 1 h of incubation and several steps of washing with 1× PBS and MilliQ water, captured IgA was eluted with 100 mM formic acid. Samples were then dried in a speedvac at 60 °C for 2.5 h and resuspended in 25 mM ammonium bicarbonate for the subsequent reduction and alkylation. For the nano-LC–ESI–MS of human IgG, human IgA and mouse IgG *N*-glycopeptides, purified tryptic human IgG, human IgA and mouse IgG glycopeptides were analyzed on a Waters Acquity M Class UPLC system. The detected glycopeptides are presented in Supplementary Table 2 and in accordance with previous reports[44–46]. Detailed methodology can be found in Supplementary Information.

### Degalactosylation of human IgGs and dot blot analysis

IgGs from each individual were incubated with β1,4-galactosidase S (New England BioLabs), according to manufacturer's instructions. The efficiency of sugar removal was verified by lectin dot blot. Briefly, 100 ng of IgG was placed on a nitrocellulose membrane and incubated for 1 h at room temperature. After, nonspecific sites were blocked with 4% bovine serum albumin (BSA) in PBS and 0.05% Tween 20 for 1 h before incubation with biotinylated ECA (diluted in blocking buffer with 0.1 mM CaCl₂; Vector Laboratories) for 1 h. After lectin incubation, the membrane was incubated with streptavidin conjugated with horseradish peroxidase (HRP) for 30 min. All incubations were performed at room temperature. Dots were then detected using ECL reagent (GE Healthcare, Life Sciences).

### IgG aggregation assay

IgG aggregation was assessed using a protein aggregation assay kit (Abcam), according to manufacturer's instructions. Detailed methodology can be found in Supplementary Information.

### Isolation of PBMCs

PBMCs were obtained from buffy coats from healthy donors as well as peripheral blood from inaugural individuals with CD (collected at the time of diagnosis) and first-degree relatives of individuals with CD. The separation of PBMCs was achieved by density centrifugation using Lymphoprep (density of 1.077 g ml⁻¹; Stem Cell Technologies). Anticoagulated blood was diluted in PBS (ratio of 1:1), slowly layered onto Lymphoprep (ratio of 1:2) and centrifuged on a gradient at 900$g$ for 30 min at room temperature. After centrifugation, the interface containing PBMCs was transferred to a conical centrifuge tube and washed with PBS. For isolation of natural killer cells, detailed methodology is provided in Supplementary Information.

### Isolation of CD14⁺ cells and generation of monocyte-derived DCs

CD14⁺ monocytes were purified from PBMCs from healthy donors using an immunomagnetic positive selection bead kit (Miltenyi Biotec) conjugated with a monoclonal antibody to human CD14 (isotype mouse IgG2a), according to the manufacturer's instructions.

CD14⁺ cells were then cultured in six-well cell culture plates in culture medium comprising RPMI-1640 and HEPES (Thermo Fisher Scientific) supplemented with 10% fetal bovine serum (FBS), 1% penicillin/streptomycin, 0.1% gentamicin, 50 ng ml⁻¹ human GM-CSF and 50 ng ml⁻¹ human IL-4 (Peprotech) for 7 days. Cells were maintained at 37 °C in a humidified incubator with 5% CO₂, and medium was replenished every 3 days.

### IgG-dependent immune cell activation

Ninety-six-well Nunc MaxiSorp high-binding plates were precoated with mannan from *S. cerevisiae* (Sigma-Aldrich) at 10 μg ml⁻¹ in 50 mM carbonate–bicarbonate coating buffer (pH 9.6) overnight at 4 °C. Afterward, a blocking step with 5% BSA was performed. Total IgGs (6 μg ml⁻¹) were added to plates previously coated with mannan, β-glucan or di-GlcNAc (10 μg ml⁻¹) for 90 min at 37 °C in a humidified incubator with 5% CO₂. To assess the effect of galactosylation, 6 μg ml⁻¹ total IgGs treated with β-gal was also used. Monocyte-derived DCs were suspended in complete RPMI medium and plated in anti-mannan IgG-coated wells at 1.5 × 10⁵ cells per well for 6 h. Additionally, for functional blockade of FcγRIIa, cells were preincubated for 1 h at 37 °C with 5 μg ml⁻¹ InVivoMab anti-human CD32a (FcγRIIA) blocking antibody (Clone IV.3) before incubation in anti-mannan IgG-coated plates. After incubation, the culture supernatants were collected and frozen at −80 °C for further cytokine analysis. Cells were washed with FACS buffer and stained for viability and surface markers (Supplementary Table 6). All incubations were performed for 30 min at 4 °C. Samples were fixed in 2% paraformaldehyde (Sigma-Aldrich), and cells were acquired by flow cytometry on a FACSCanto II system using FACSDiva software. Data were analyzed using FlowJo version 10.5.3 (Tree Star) using the gating strategy shown in Supplementary Fig. 9a. For IgG-dependent NK cell degranulation, detailed methodology is provided in Supplementary Information, and flow cytometry data analysis was performed using the gating strategy shown in Supplementary Fig. 9b.

## Plasma cell characterization in human PBMCs

PBMCs from inaugural individuals with CD (collected at the time of diagnosis), first-degree relatives of individuals with CD and HC individuals were collected, and plasma cell characterization was performed by flow cytometry using surface and intracellular markers (Supplementary Table 6), as described above. Cells were acquired by flow cytometry on a Cytek Aurora system with SpectraFlow v3.2.1, and data were analyzed in FlowJo version 10.5.3 (Tree Star) using the gating strategy shown in Supplementary Fig. 9c.

## Cytokine quantification

NK cells were cultured in a 96-well Nunc MaxiSorp high-binding plate precoated with mannan and total IgGs at $2 \times 10^5$ cells per well for 48 h at 37 °C in a humidified incubator with 5% $CO_2$. IFNγ levels in the supernatants were quantified by ELISA using a human IFNγ DuoSet ELISA (R&D Systems), according to the manufacturer's instructions.

Supernatants from 6-h DC cultures were analyzed by flow cytometry (BD Accuri C6 Plus) using a BD Cytometric Bead Array (CBA) Human Inflammatory Cytokine kit (BD Biosciences), following the manufacturer's instructions.

For mouse experiments, TNF and IL-1β levels were measured in supernatants from colonic explants (in culture for 24 h) by ELISA using commercially available kits (Invitrogen) according to the manufacturer's instructions. Levels of cytokines (measured in pg ml$^{-1}$) were normalized by explant weight (g).

## Bulk RNA-sequencing data processing and analysis

Sequenced reads were aligned to the human GRCh38 (Ensembl version 94) reference genome using HISAT2 (v2.0.5) with default settings. Counts per gene were generated from the alignment files by FeatureCounts (v1.5.0-p3). Differential expression analysis between pairwise conditions was conducted in R (v4.1.2) using DESeq2 (v1.34). For each comparison, log$_2$ (fold change) results were shrunken using the apeglm (v1.23.1) package to remove noise (differentially expressed genes with low counts and/or high dispersion values) while preserving significant differences.

Overrepresentation pathway analysis was conducted in clusterProfiler (v4.11.0.001) to the human KEGG ontology database for the set of upregulated and downregulated genes (false discovery rate ≤ 0.05) separately. Pathways with an adjusted P value of <0.05 were regarded as significant. Bubble diagrams were constructed for the visualization of the top six enriched pathways, after excluding those belonging to human diseases.

A heat map of the differentially expressed genes between conditions for the top six upregulated KEGG pathways was plotted using complexHeatmap (v2.10.0). Volcano plots representing differentially expressed genes were generated using ggplot2 (v3.4.0). Data were deposited in the European Nucleotide Archive with the dataset identifier PRJEB76671.

## C1q (complement) binding assay

C1q binding was assessed by incubation with complement component C1q to IgGs added to plates previously coated with mannan (10 µg ml$^{-1}$) for 90 min at 37 °C in a humidified incubator with 5% $CO_2$ and detected with HRP sheep anti-human C1q. Detailed methodology can be found in Supplementary Information.

## Gene expression analysis using quantitative PCR with reverse transcription

Total RNA from DCs was extracted using TRIzol reagent (Thermo Fisher Scientific). Total RNA was quantified and retrotranscribed to cDNA using Superscript IV Reverse Transcriptase (Invitrogen), according to the instructions of the kit. cDNA was amplified using Taqman Universal PCR Master Mix (Applied Biosystems) and Taqman probes (Supplementary Table 7). Amplification data were acquired with a 7500 Fast Real-Time PCR System (Applied Biosystems); 18S was used as the housekeeping gene. Relative quantification values for gene expression were calculated based on the change in cycling threshold ($\Delta C_t$) method using the following equation: $2^{-(\text{target gene mRNA expression} - \text{housekeeping gene mRNA expression})}$.

## Phospho-NF-κB immunoblotting

After 3 h of culture with IgGs, DCs were lysed in RIPA buffer (150 mM sodium chloride, 50 mM Tris-HCl, 1% Nonidet P-40, 0.5% sodium deoxycholate and 10% SDS) with 1% sodium orthovanadate (Sigma), 1% phenylmethylsulfonyl fluoride (Thermo Fisher Scientific) and 4% cOmplete protease inhibitor cocktail (Roche). Protein lysates were separated on 10% resolving polyacrylamide gels. Proteins were then transferred to nitrocellulose membranes (GE Healthcare, Amersham) and blocked with 4% BSA in Tris-buffered saline with 0.1% Tween 20 (TBS-T) for 1 h. Membranes were incubated with rabbit anti-human phosphorylated NF-κB (93H1, Cell Signaling Technology) diluted 1:1,000 in 4% nonfat dry milk in TBS-T. For protein loading (control), membranes were blocked with 4% nonfat dry milk for 1 h and incubated with mouse anti-human vinculin (V284, Santa Cruz Biotechnology) diluted 1:500 in 1% nonfat dry milk in TBS-T. Bound anti-phospho-NF-κB and anti-vinculin were detected using goat anti-rabbit IgG HRP and chicken anti-mouse IgG HRP, respectively. Chemiluminescence was determined using ECL prime (GE Healthcare, Amersham).

## Animal model of colitis

All mouse procedures were approved by the Institute for Research and Innovation in Health (i3S) Animal Ethics Committee for animal experimentation under Portuguese regulations (DGAV license number 009268/2022-06-02). Mice were housed at the Association for Assessment and Accreditation of Laboratorial Animal Care-accredited i3S animal facility in a temperature-controlled (20–24 °C) room maintained at a humidity of 45–55% under a 12-h light/12-h dark period.

C57BL/6 WT mice aged between 6 and 10 weeks were inoculated with 12 mg of mannan subcutaneously every 3 days over 5 weeks. The control group received PBS by the same route at similar time points. Levels of ASCA were monitored after 5 weeks by collecting a small amount of blood by the tail vein and analyzing by ELISA (QUANTA Lite ASCA IgG ELISA, QUANTA; goat anti-mouse IgG HRP (Bethyl) was used as the detection antibody). Sera from mannan-treated and control mice were collected upon cardiac puncture at the experimental endpoint, and total IgGs were isolated and quantified (as previously described for human samples).

In vivo intestinal permeability was assessed by administration of fluorescein isothiocyanate-labeled dextran. Food and water were withdrawn for 8 h. Mice were administered 44 mg per 100 g (body weight) fluorescein isothiocyanate-labeled dextran (TdB Consultancy; 4 kDa) by oral gavage. Serum was collected from the tail vein 4 h later, and fluorescence intensity was measured by spectrophotofluorimetry (excitation: 485 nm; emission: 528 nm).

WT and FcγR-KO (C57BL/6 background) mice aged 10–16 weeks (recipient) were intraperitoneally injected with 100 µg of ASCA-enriched IgG pool or, as a control, 100 µg of PBS-derived IgGs on days 0 and 3. At day 0, colitis was induced chemically by treatment with 2% DSS (36,000–50,000 Da; MP Biomedicals) via the drinking water (provided ad libitum) from day 0 to day 7, which was then switched to tap water. Mice were monitored daily, and the DAI was assessed according to the parameters specified in Supplementary Table 5. Animals were killed according to the humane endpoints.

For oral immunization with mannan, mice were treated with 12 mg of mannan by oral gavage in the first 3 days of each week over 3 weeks (total of nine administrations), followed by DSS-induced colitis.

To isolate lamina propria leukocytes, colonic fragments 0.5–1 cm in size were incubated in DMEM supplemented with 1 mM $CaCl_2$, 1 mM $MgCl_2$, 1.5 mg ml$^{-1}$ collagenase IV (Sigma) and 0.4 mg ml$^{-1}$ dispase (Gibco) under agitation at 100 rpm and 37 °C for 40 min. Tissues were dissociated and filtered through a 70-µm cell strainer (BD Biosciences).

Cell suspensions were resuspended in RPMI-1640 medium supplemented with 10% FBS and 1% penicillin/streptomycin and layered on Lymphoprep solution at a ratio of 1:2 (Lymphoprep:cell suspension). After gradient centrifugation at 800*g* for 20 min at 20 °C (without acceleration or break), immune cells (retained in the interface) were collected for staining using the antibodies listed in Supplementary Table 6. Cells were acquired by flow cytometry on a Cytek Aurora system with SpectraFlow v3.2.1, and data were analyzed in FlowJo version 10.5.3 (Tree Star) using the gating strategy shown in Supplementary Fig. 9d.

### BMDCs and coculture with mouse ASCA IgGs
Bone marrow was extracted from the femurs and tibias of C57BL/6 WT mice, and isolated cells were then cultured in RPMI supplemented with 10% FBS, penicillin/streptomycin (100 U ml⁻¹) and mouse recombinant proteins IL-4 and GM-CSF (50 ng ml⁻¹; *Escherichia coli*, PeproTech). On the third day of culture, the medium was replaced, and the cells were cultured for additional 4 days. Differentiated DCs were then incubated with mouse ASCA IgGs and PBS IgGs (control), similar to the protocol described for IgG-dependent human DC activation. Cells were acquired by flow cytometry on a Cytek Aurora system with SpectraFlow v3.2.1, and data were analyzed using FlowJo version 10.5.3 (Tree Star).

### Statistical analysis
Data were obtained from at least two independent experiments and processed using GraphPad Prism 10.2.3 (GraphPad Software). Statistical significance was determined by Kruskal–Wallis test, ANOVA, unpaired *t*-test or Mann–Whitney test, according to normality of the samples. For animal experiments, a two-way ANOVA was applied to the analysis of DAI. The robust regression and outlier removal test was performed to identify outliers, and those values were eliminated from the analysis. Flow cytometry data were analyzed using FlowJo v10. The number of samples is indicated in each figure legend.

### Batch correction of glycan traits.
Glycome profiles for 2,991 samples were obtained using 40 plates. Batch correction across different plates was performed using the Combat function from the 'sva' package available in R Cran[47].

### Association between glycan traits and disease onset.
The association between each glycan marker and disease onset was assessed via logistic regression after adjusting for age, race and sex. Logistic regression was estimated for each time point before diagnosis separately. *P* values from logistic regressions were adjusted for multiple comparisons via the Benjamini–Hochberg method[48], and only markers with an adjusted *P* value smaller than 10% were reported as significant.

### Predictive performance.
The predictive performance of different glycan traits was evaluated based on leave-one-out cross-validation. Specifically, for each sample, we used the remaining $n - 1$ samples to fit a logistic regression where the disease status (that is, HC versus CD and HC versus UC) was modeled as a function of each glycan trait, sex, race and age. The estimated model was then used to predict disease status of the left-out sample. The predictive performance of each glycan was assessed via area under the receiver operating characteristic curve (ROC). Predictive performance was evaluated for each time point before diagnosis separately. The R package pROC was used to compute area under the ROC values and to plot the ROC[49].

### Coexpression network analysis.
We performed coexpression network analysis to identify the associations across glycan traits and antimicrobial antibodies for individuals with CD and HC individuals jointly via joint random forest[50]. Coexpression networks were estimated at the furthest time point before diagnosis. Permutation-based techniques were used to find associations significant at a false discovery rate of 5% (ref. 50).

### Deriving cell-type score from proteomic data.
Based on prior knowledge of cell-type markers, we leveraged the LM22 signature matrix including the expression of different genes for 22 cell types[24]. Leveraging cell-type signatures from this study, we computed single-sample gene set enrichment analysis via the R package GSVA[51] (method = 'zscore') using proteomic data available for different PRE-DICT samples. We then correlated the estimated cell-type score with CD/HC status via the Wilcoxon rank-sum test.

All analyses were performed using R statistical software (version 3.6.3, R Foundation for Statistical Computing).

### Figure preparation
Some schematic figures were drawn using pictures from Servier Medical Art, licensed under a Creative Commons Attribution 4.0 Unported License (https://creativecommons.org/licenses/by/4.0/).

### Reporting summary
Further information on research design is available in the Nature Portfolio Reporting Summary linked to this article.

## Data availability
The RNA-sequencing data have been deposited in the EMBL Nucleotide Sequence Database, European Nucleotide Archive, with the dataset identifier PRJEB76671. Participant-related data may be subject to donor confidentiality. The data displayed in Fig. 1 and Supplementary Figs. 1a–c, 2 and 5a–c are available from the corresponding author upon request. Reagents, materials and protocols are available from the corresponding author upon request. Source data are provided with this paper.

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

## Acknowledgements

We wish to acknowledge the Gastroenterology Department of Centro Hospitalar Universitário de Santo António, in particular P. Lago, for providing samples from individuals with established CD. We kindly thank J. Rojo from the Instituto de Investigaciones Químicas (Universidad de Sevilla) for providing us with the di-GlcNAc glycodendrimer. We would also like to acknowledge J. V. Ravetch (Rockefeller University) and M. S. Cragg (University of Southampton) for kindly providing us with the FcγR-deficient mice used for the in vivo studies. S.S.P. acknowledges funding from the US Department of Defense, US Army Medical Research Acquisition Activity and FY18

Peer-Reviewed Medical Research Program Investigator-Initiated Research Award (award number W81XWH1920053). S.S.P. also acknowledges funding from the European Crohn's and Colitis Organisation (ECCO) Pioneer Award 2021, the International Organization for the study of Inflammatory Bowel Disease (IOIBD) and the Portuguese Foundation for Science and Technology (FCT; EXPL/MED-ONC/0496/2021). J.G. acknowledges funding from European Society of Clinical Microbiology and Infectious Diseases (ESCMID Research Grant 2022), European Crohn's and Colitis Organisation (ECCO Grant 2023) and FCT (2020.00088.CEECIND). C.S.R. thanks FCT for funding (2020.08422.BD). I.A. acknowledges funding from FCT (2022.00337.CEECIND) and the BIAL Foundation and Portuguese Medical Association (Maria de Sousa Award 2023). E.L.-G. thanks FCT for funding (UI/BD/152866/2022). F.P. and Z.H.G. were partially supported by the Kenneth-Rainin Foundation (20210021). B.C. acknowledges funding from FCT (CEECINST/00123/2021/CP1772/CT0001). J.T. acknowledges funding from the Portuguese Society of Gastroenterology and from Luz Saúde (Grupo dE iNvestIgação em Patologia Digestiva LUz Saúde LH.INV.F2019015). This study was also cofunded by the European Union (GlycanTrigger, grant agreement number 101093997). The views and opinions expressed are, however, those of the author(s) only and do not necessarily reflect those of the European Union or the European Research Council Executive Agency. Neither the European Union nor the granting authority can be held responsible for them. This study was conducted under support of Peer-Reviewed Medical Research Program (PR180831P1). The views expressed in this article reflect the results of the research conducted by the authors and do not necessarily reflect the official policy or position of the Department of the Navy, Department of Defense, the Henry M. Jackson Foundation for the Advancement of Military Medicine or the US Government. There are no restrictions on its use. This article was prepared while R.M.L. was employed at Henry M. Jackson Foundation for the Advancement of Military Medicine. The opinions expressed in this article are those of the authors and do not reflect the view of the National Institutes of Health, the Department of Health and Human Services or the US Government. C.K.P. is an employee of the US Government. This work was prepared as part of official duties. Title 17 U.S.C. §105 provides that 'Copyright protection under this title is not available for any work of the United States Government'. Title 17 U.S.C. §101 defines a US Government work as a work prepared by a military service member or employee of the US Government as part of that person's official duties.

## Author contributions

J.G. and C.S.R. designed the study, performed the experiments and data analysis and wrote the paper. F.P., B.T. and Z.H.G. performed the bioinformatic analyses on IgG glycome data. I.A. and E.L.-G. designed, performed and analyzed the animal experiments. B.C. analyzed the RNA-sequencing data. A.M.D. contributed to in vitro experiments. C.M.-B. and J.R. collected human biological samples and clinical data from first-degree relatives and inaugural CD cases. R.M.L. prepared biological samples and demographic data from the PREDICTS cohort. M.N., J.Š. and S.H. performed the IgG/IgA glycome characterizations. R.U. contributed in scientific discussions. J.T., G.L., J.-F.C. and C.K.P. participated in the discussion of the study design and contributed to critical revisions of the paper. S.S.P. designed the study, acquired funding and wrote the paper.

## Competing interests

G.L. is the founder and owner of Genos, Ltd., a private research organization that specializes in high-throughput glycomic analysis, and has several patents in this field. M.N. and J.Š. are employees of Genos, Ltd. The other authors declare no competing interests.

## Additional information

**Correspondence and requests for materials** should be addressed to Salomé S. Pinho.

# Reporting Summary

## Statistics

For all statistical analyses, confirm that the following items are present in the figure legend, table legend, main text, or Methods section.

| n/a | Confirmed | |
|---|---|---|
| ☐ | ☒ | The exact sample size (*n*) for each experimental group/condition, given as a discrete number and unit of measurement |
| ☐ | ☒ | A statement on whether measurements were taken from distinct samples or whether the same sample was measured repeatedly |
| ☐ | ☒ | The statistical test(s) used AND whether they are one- or two-sided *Only common tests should be described solely by name; describe more complex techniques in the Methods section.* |
| ☐ | ☒ | A description of all covariates tested |
| ☐ | ☒ | A description of any assumptions or corrections, such as tests of normality and adjustment for multiple comparisons |
| ☐ | ☒ | A full description of the statistical parameters including central tendency (e.g. means) or other basic estimates (e.g. regression coefficient) AND variation (e.g. standard deviation) or associated estimates of uncertainty (e.g. confidence intervals) |
| ☐ | ☒ | For null hypothesis testing, the test statistic (e.g. *F*, *t*, *r*) with confidence intervals, effect sizes, degrees of freedom and *P* value noted *Give P values as exact values whenever suitable.* |
| ☒ | ☐ | For Bayesian analysis, information on the choice of priors and Markov chain Monte Carlo settings |
| ☒ | ☐ | For hierarchical and complex designs, identification of the appropriate level for tests and full reporting of outcomes |
| ☒ | ☐ | Estimates of effect sizes (e.g. Cohen's *d*, Pearson's *r*), indicating how they were calculated |

*Our web collection on statistics for biologists contains articles on many of the points above.*

## Software and code

Policy information about availability of computer code

| Data collection | FACSDiva™ V6.1.3 software; SpectraFlow v3.2.1; MassLynx software 4.1; HyStar software version 4.1 |
|---|---|
| Data analysis | FlowJo version 10.5.3; FeatureCounts (v1.5.0-p3); R statistical software (version 3.6.3 and v4.1.2); DESeq2 (v1.34); apeglm (v1.23.1); clusterProfiler (v4.11.0.001); complexHeatmap (v2.10.0); ggplot2 (v3.4.0); GraphPad Prism 10.2.3 |

For manuscripts utilizing custom algorithms or software that are central to the research but not yet described in published literature, software must be made available to editors and reviewers. We strongly encourage code deposition in a community repository (e.g. GitHub). See the Nature Portfolio guidelines for submitting code & software for further information.

## Data

Policy information about availability of data

All manuscripts must include a data availability statement. This statement should provide the following information, where applicable:
- Accession codes, unique identifiers, or web links for publicly available datasets
- A description of any restrictions on data availability
- For clinical datasets or third party data, please ensure that the statement adheres to our policy

The RNA sequencing data have been deposited in the EMBL Nucleotide Sequence Database – European Nucleotide Archive (ENA) with the dataset identifier PRJEB76671. Patients-related data may be subject to confidentiality. Data displayed in Figure 1, Supplementary Figures 1a, 1b and 1c, Supplementary Figure 2, and

Supplementary Figures 5a, 5b and 5c are available from the corresponding author upon request. The source data for remaining Figures and Supplementary Figures are provided as a Source Data file. Reagents, materials and protocols are available upon request to the corresponding author.

## Research involving human participants, their data, or biological material

Policy information about studies with human participants or human data. See also policy information about sex, gender (identity/presentation), and sexual orientation and race, ethnicity and racism.

| | |
|---|---|
| Reporting on sex and gender | We use the term "sex" consistently throughout the manuscript. All information regarding donors is listed on supplementary table 1, 3 and 4. |
| Reporting on race, ethnicity, or other socially relevant groupings | IgG glycome analysis was performed by two-sided t-test from logistic regression model (-log10 scale) after adjusting for sex, race and age. |
| Population characteristics | For the PREDICTS cohort, a total of 750 individuals were recruited (CD: n=251, 81.64% males; HC: n=250, 91.68% males; UC: n=249, 89.72% males), comprising a total of 2991 serum samples. The mean age was: CD-31.70 years; HC-29.67 years; UC-29.77 years.<br>The established CD cohort was composed by 10 individuals (70% males) comprising 33 serum samples. The mean age was 53.50 years, and the mean time since diagnosis was 19.04 years. Information on clinical activity or treatments is presented in supplementary information.<br>The cohort for first-degree relatives (FDR) and inaugural CD patients was composed by: 6 inaugural CD (83% male); 7 FDR (29% male) and 6 healthy control individuals (17% males). The mean age was 32 years for inaugural CD, 26 years for FDR and 27 for healthy individuals. |
| Recruitment | Samples were retrieved from the US Department of Defense Serum Repository (DoDSR) or recruited in the local hospitals. Only patients able to give their informed consent were recruited. We cannot identify any specific bias in the process that may significantly influence the results. |
| Ethics oversight | This study was approved as non-human subjects' research (PJT 19-08) by the Naval Medical Research Command, Silver Spring, MD. Ethical approval was obtained at the Ethical Committees of all Hospitals. |

Note that full information on the approval of the study protocol must also be provided in the manuscript.

# Field-specific reporting

Please select the one below that is the best fit for your research. If you are not sure, read the appropriate sections before making your selection.

☒ Life sciences ☐ Behavioural & social sciences ☐ Ecological, evolutionary & environmental sciences

For a reference copy of the document with all sections, see nature.com/documents/nr-reporting-summary-flat.pdf

# Life sciences study design

All studies must disclose on these points even when the disclosure is negative.

| | |
|---|---|
| Sample size | This study is focused on the exploitation of altered immune-mediated mechanisms in the context of health to inflammation transition. We believe that the final sample size is enough considering that all sub-groups of age,sex, disease severity are represented by a reasonable number of individuals and conditions, as well as based on previous evidence. By adding a control group with a significant dimension, we also add reliability to our results. |
| Data exclusions | No data were excluded from the analysis. |
| Replication | All the assays involved testing independent clinical samples from patients and controls. For animal experimentation, biological replicates were considered in the analysis. The assay with BMDCs was performed once, but with four technical replicates. |
| Randomization | Randomization was taken into consideration in serum IgG glycome profile. The groups for comparison were established according to the clinical status of the patients/individuals, and the analysis was performed on that basis. |
| Blinding | The investigators were blinded to group allocation during data collection and analysis. A code was assigned to each participant and the data collection and analysis was performed without the knowledge of which group they appertain. |

# Reporting for specific materials, systems and methods

We require information from authors about some types of materials, experimental systems and methods used in many studies. Here, indicate whether each material, system or method listed is relevant to your study. If you are not sure if a list item applies to your research, read the appropriate section before selecting a response.

## Materials & experimental systems

| n/a | Involved in the study |
|---|---|
| ☐ | ☒ Antibodies |
| ☒ | ☐ Eukaryotic cell lines |
| ☒ | ☐ Palaeontology and archaeology |
| ☐ | ☒ Animals and other organisms |
| ☒ | ☐ Clinical data |
| ☒ | ☐ Dual use research of concern |
| ☐ | ☐ Plants |

## Methods

| n/a | Involved in the study |
|---|---|
| ☒ | ☐ ChIP-seq |
| ☐ | ☒ Flow cytometry |
| ☒ | ☐ MRI-based neuroimaging |

# Antibodies

| Antibodies used | |
|---|---|
| | 1. Brilliant Violet 510 anti-human CD3, clone OKT3, 317332, BioLegend |
| | 2. PE anti-human CD56, clone HCD56, 318306, BioLegend |
| | 3. APC anti-human Granzyme B, MHGB05, Invitrogen |
| | 4. FITC anti-human CD107a (LAMP-1), 328606, BioLegend |
| | 5. PE anti-human CD14, clone 61D3, 12-0149-42, Invitrogen |
| | 6. APC anti-human CD11c, clone B015, 17-0128-42, Invitrogen |
| | 7. PE-Cyanine5 anti-human CD86, clone IT2.2, 15-0869-42, Invitrogen |
| | 8. Rabbit anti-human CD209, AHP627, Bio-Rad |
| | 9. Polyclonal Swine anti rabbit Immunoglobulins/FITC, F0205, Dako |
| | 10. Anti-hDectin2 Affinity Purified Goat IgG, AF3114, R&D Systems |
| | 11. Polyclonal Rabbit anti-Goat Immunoglobulins/Biotinylated, E0466, Dako |
| | 12. Streptavidin PE-Cyanine7, 25-4317-82, Invitrogen |
| | 13. PE-CF594 anti-human CD138, clone MI15, 564606, BD Biosciences |
| | 14. PE-Cyanine5 anti-human CD38, clone HIT2, 303507, BioLegend |
| | 15. BV605 anti-human CD3, clone 17A2, 100237, BioLegend |
| | 16. cFR685 anti-human CD19, clone HIB19, R7-20118, Cytek |
| | 17. BB700 anti-human IgG, clone G18-145, 742235, BD Biosciences |
| | 18. PE-Cyanine7 anti-human IgM, clone MHM-88, 314531, BioLegend |
| | 19. BV605 anti-mouse CD45, clone 30-F11, 103139, BioLegend |
| | 20. eF450 anti-mouse CD11c, clone N418, 48-0114-82, Invitrogen |
| | 21. PE anti-mouse NKp46, clone 29A1.4, 12-3351-80, Invitrogen |
| | 22. PerCP-Cy5.5 anti-mouse CD3, 100328, BioLegend |
| | 23. PE-Cy5 anti-mouse MHCII, clone M5/114.15.2, 15-5321-81, Invitrogen |
| | 24. PE-Cy7 anti-mouse CD45, clone 30-F11, 25-0451-82, Invitrogen |
| | 25. FITC anti-mouse IFNg, clone XMG1.2, 11-7311-82, Invitrogen |
| | 26. PE-Cy7 anti-mouse CD16/32, clone 93, 101318, BioLegend |
| | 27. Purified anti-mouse CD16/32, clone 93, 101302, BioLegend |

| Validation | |
|---|---|
| | 1. https://www.biolegend.com/en-us/products/brilliant-violet-510-anti-human-cd3-antibody-8009 |
| | 2. https://www.biolegend.com/en-us/products/pe-anti-human-cd56-ncam-antibody-3796 |
| | 3. https://www.thermofisher.com/antibody/product/Granzyme-B-Antibody-clone-GB12-Monoclonal/MHGB05 |
| | 4. https://www.biolegend.com/en-us/products/fitc-anti-human-cd107a-lamp-1-antibody-4966 |
| | 5. https://www.thermofisher.com/antibody/product/CD14-Antibody-clone-61D3-Monoclonal/12-0149-42 |
| | 6. https://www.thermofisher.com/antibody/product/CD11c-Antibody-clone-BU15-Monoclonal/17-0128-42 |
| | 7. https://www.thermofisher.com/antibody/product/CD86-B7-2-Antibody-clone-IT2-2-Monoclonal/15-0869-42 |
| | 8. https://www.bio-rad-antibodies.com/polyclonal/human-cd209-antibody-ahp627.html?f=purified |
| | 9. https://www.agilent.com/store/pt_BR/Prod-F020502-2/F020502-2 |
| | 10. https://www.rndsystems.com/products/human-dectin-2-clec6a-antibody_af3114 |
| | 11. https://www.agilent.com/en/oem-polyclonal-antibodies |
| | 12. https://www.thermofisher.com/order/catalog/product/25-4317-82 |
| | 13. https://www.bdbiosciences.com/en-au/products/reagents/flow-cytometry-reagents/research-reagents/single-color-antibodies-ruo/pe-cf594-mouse-anti-human-cd138.564606 |
| | 14. https://www.biolegend.com/en-us/products/pe-cyanine5-anti-human-cd38-antibody-747 |
| | 15. https://www.biolegend.com/en-us/products/brilliant-violet-605-anti-mouse-cd3-antibody-8503 |
| | 16. https://welcome.cytekbio.com/hubfs/TDS%20cFluor%20Reagents/R8-50117%20Rev.B_TDS_cFluor%20R685%20huCD19%20(HIB19).pdf |
| | 17. https://www.bdbiosciences.com/en-ie/products/reagents/flow-cytometry-reagents/research-reagents/single-color-antibodies-ruo/bb700-mouse-anti-human-igg.742235 |
| | 18. https://www.biolegend.com/en-us/products/pe-cyanine7-anti-human-igm-antibody-12467 |
| | 19. https://www.biolegend.com/en-us/products/brilliant-violet-605-anti-mouse-cd45-antibody-8721 |
| | 20. https://www.thermofisher.com/antibody/product/CD11c-Antibody-clone-N418-Monoclonal/48-0114-82 |
| | 21. https://www.thermofisher.com/antibody/product/CD335-NKp46-Antibody-clone-29A1-4-Monoclonal/12-3351-80 |
| | 22. https://www.biolegend.com/en-us/products/percp-cyanine5-5-anti-mouse-cd3epsilon-antibody-4191 |
| | 23. https://www.thermofisher.com/antibody/product/MHC-Class-II-I-A-I-E-Antibody-clone-M5-114-15-2-Monoclonal/15-5321-81 |
| | 24. https://www.thermofisher.com/antibody/product/CD45-Antibody-clone-30-F11-Monoclonal/25-0451-82 |
| | 25. https://www.thermofisher.com/antibody/product/IFN-gamma-Antibody-clone-XMG1-2-Monoclonal/11-7311-82 |

26.    https://www.biolegend.com/en-us/products/pe-cyanine7-anti-mouse-cd16-32-antibody-6355
27.    https://www.biolegend.com/en-us/products/purified-anti-mouse-cd16-32-antibody-190

# Animals and other research organisms

Policy information about studies involving animals; ARRIVE guidelines recommended for reporting animal research, and Sex and Gender in Research

| | |
|---|---|
| Laboratory animals | C57BL/6 were used as WT mice. FcgR KO mice (C57BL/6 background) were kindly provided by Jeffrey V. Ravetch (Rockefeller University, New York, USA) and Mark S. Cragg (University of Southampton, United Kingdom). Mice were housed at the AAALAC-accredited i3S animal facility in a controlled-temperature room (20-24ºC), 45-55% humidity, and under a 12-hour light/12-hour dark period. WT mice were used between 6-13 weeks of age. FcgR KO mice were used between 10-13 weeks of age. |
| Wild animals | No wild animals were used in the study. |
| Reporting on sex | For mannan inoculation and collection, both female and male mice were used. Since the goal was to collect mannan-specific IgGs (or the respective PBS IgGs from the control), sex is not expected to act as a confounding factor. For DSS experiments only with WT animals (Fig 4b), females were used to reduce possible sex-dependent variability, taking into consideration our previous evidence. For experiments with FcgR KO mice, both males and females were used; aged- and sex-matched WT were used as controls; no major variability was found. |
| Field-collected samples | N/A |
| Ethics oversight | All mouse procedures were approved by the Institute for Research and Innovation in Health (i3S) Animal Ethics Committee for animal experimentation under Portuguese regulations (DGAV license number 009268/2022-06-02). |

Note that full information on the approval of the study protocol must also be provided in the manuscript.

# Plants

| | |
|---|---|
| Seed stocks | N/A |
| Novel plant genotypes | N/A |
| Authentication | N/A |

# Flow Cytometry

## Plots

Confirm that:

☒ The axis labels state the marker and fluorochrome used (e.g. CD4-FITC).

☒ The axis scales are clearly visible. Include numbers along axes only for bottom left plot of group (a 'group' is an analysis of identical markers).

☒ All plots are contour plots with outliers or pseudocolor plots.

☒ A numerical value for number of cells or percentage (with statistics) is provided.

## Methodology

| | |
|---|---|
| Sample preparation | PMBCs were obtained from buffy coats from healthy donors, as well as peripheral blood from inaugural CD patients (collected at the time of diagnosis) and first-degree relatives of CD patients. Anticoagulated blood was diluted in PBS (ratio 1:1), slowly layered onto Lymphoprep™ (ratio 1:2) and centrifuged by gradient at 900 x g for 30 minutes at room temperature. After centrifugation, the interface containing PBMCs was transferred to a conical centrifuge tube and washed with PBS. PBMCs from inaugural CD patients (collected at the time of diagnosis), first-degree relatives of CD patients, and healthy controls were collected and the plasma cell characterization was performed by flow cytometry, using surface and intracellular markers. Cells were fixed and permeabilized with Transcription Factor Staining Buffer Set (eBioscience), followed by intracellular staining.<br>For DC culture, after 6h cells were washed with FACS buffer and stained for viability and surface markers. All incubations were performed for 30 minutes at 4°C. Samples were fixed by 2% of paraformaldehyde (Sigma-Aldrich).<br>For NK culture, following 1 hour of culture, 10 μg/ml of Brefeldin A and GolgiStop™ Protein Transport Inhibitor were added and incubated 5 hours in similar conditions. After incubation, cells were stained for viability, as well as for surface markers. Cells were then fixed and permeabilized with Transcription Factor Staining Buffer Set (eBioscience), followed by intracellular |

staining.

In in vivo experiments, lamina propria leukocytes were isolated from colonic fragments incubated in DMEM medium supplemented with 1mM CaCl2, 1mM MgCl2, 1.5 mg/mL of collagenase IV (Sigma), and 0.4 mg/mL of dispase (Gibco), under 100 rpm agitation at 37ºC for 40 minutes. Tissues were dissociated and filtered through a 70 μm cell strainer (BD Biosciences). Cell suspension was resuspended in RPMI 1640 medium supplemented with 10% FBS, and 1% penicillin/streptomycin, and layered upon Lymphoprep solution in a proportion of 1:2 (Lymphoprep:cell suspension). After gradient centrifugation at 800g for 20 minutes at 20ºC (without acceleration or break), immune cells (retained in the interface) were collected for the staining.

| Instrument | FACSCanto™ II system (BD Biosciences); Cytek Aurora™ system (Cytek). |
|---|---|
| Software | Samples were acquired using the BD FACSDiva software V6.1.3 or SpectraFlow v3.2.1, and data was analyzed using FlowJo software, version 10.5.3. |
| Cell population abundance | No cell sorting was performed in this study. |
| Gating strategy | For the co-culture of human DCs with ASCA IgGs, cells were gated by CD14low, CD11c+ cells, after exclusion of duplets and dead cells. The median fluorescence intensity (MFI) of CD86, DC-SIGN and dectin-2 was assessed in CD11c+ cells.<br>For the co-culture of human NK cells with ASCA IgGs, cells were identified as CD3-, CD56+ cells, after exclusion of duplets and dead cells. The median fluorescence intensity (MFI) of granzyme B and CD107a (LAMP-1) was assessed in CD3-CD56+ cells.<br>For the analysis of B cells, plasma cells and plasmablasts, duplets and dead cells were excluded and cell populations were defined as follows: B cells - CD3-CD19+; plasma cells - CD3-CD138+CD19-CD38+; plasmablasts - CD3-CD138+CD19+.<br>For the animal model, duplets and dead cells were excluded, and DCs were identified as CD45+CD3-CD11c+ cells, while NK cells were selected as CD45+CD3-NKp46+ cells. IFNγ and granzyme B were assessed within NK cells, while MHC-II was analysed within DCs. The gating strategy is included in the manuscript in the Supplementary Information. |

☒ Tick this box to confirm that a figure exemplifying the gating strategy is provided in the Supplementary Information.

