## [Peer Review File · Nature Immunology]

A unique serum glycosylation signature of IgGs predicts the development of Crohn's disease and is associated with pathogenic anti-mannose glycan (ASCA) antibodies

Corresponding Author: Professor Salomé Pinho

Version 0:

Decision Letter:

4th Aug 2023

Dear Professor Pinho,

Thank you for providing your response to reviewers comments on your article "A unique serum glycosylation signature of IgGs predicts the development of Crohn's disease and is associated with pathogenic anti-mannose glycan (ASCA) antibodies". We would be interested in considering a revised version of the manuscript as outlined in your response.

We hope you will find the referees' comments useful as you decide how to proceed. If you wish to submit a substantially revised manuscript, please bear in mind that we will be reluctant to approach the referees again in the absence of major revisions.

Please ensure that RNAseq experiments suggested by Reviewer #1 are performed, along with the other revisions you proposed.

If you choose to revise your manuscript taking into account all reviewer and editor comments, please highlight all changes in the manuscript text file [OPTIONAL: in Microsoft Word format].

* If you have not done so already please begin to revise your manuscript so that it conforms to our Article format instructions at <http://www.nature.com/ni/authors/index.html>. Refer also to any guidelines provided in this letter.

The Reporting Summary can be found here:

Finally, please ensure that you retain unprocessed data and metadata files after publication, ideally archiving data in

perpetuity, as these may be requested during the peer review and production process or after publication if any issues arise.

Link Redacted

If you wish to submit a suitably revised manuscript we would hope to receive it within 6 months. If you cannot send it within this time, please let us know. We will be happy to consider your revision so long as nothing similar has been accepted for publication at Nature Immunology or published elsewhere.

Nature Immunology is committed to improving transparency in authorship. As part of our efforts in this direction, we are now requesting that all authors identified as 'corresponding author' on published papers create and link their Open Researcher and Contributor Identifier (ORCID) with their account on the Manuscript Tracking System (MTS), prior to acceptance. ORCID helps the scientific community achieve unambiguous attribution of all scholarly contributions. You can create and link your ORCID from the home page of the MTS by clicking on 'Modify my Springer Nature account'. For more information please visit www.springernature.com/orcid.

Thank you for the opportunity to review your work.

Sincerely,

Stephanie Houston, PhD
Senior Editor
Nature Immunology

Reviewers' Comments:

Reviewer #1:

Remarks to the Author:

The manuscript by Gaifem and colleagues addresses a timely and clinically very interesting topic. Understanding the very early phases of autoimmune diseases is critical to allow an early intervention. Ultimately this early intervention may allow resetting the immune system and cure autoimmunity. However, this requires that specific biomarkers predicting disease development become available to justify therapeutic intervention. Gaifem and colleagues study changes in IgG glycosylation in a cohort of patients with Crohn's disease (CD). In CD and more general in inflammatory bowel disease anti-saccharomyces cerevisiae antibodies (ASCA) represent a biomarker for CD. Moreover, changes in serum protein glycosylation and complement pathway activation have been associated with early CD development. The authors make use of a cohort of patient samples collected up to six years before disease development. Interestingly, an agalactosylated IgG glycoform (IgG-G0F) specifically appeared in CD patients but not in patients with UC years before disease development. More importantly, on the IgG2 background this glycovariant correlated with more severe disease development and ASCA IgG. The authors further demonstrate that ASCA isolated from patients before development of disease are able to activate monocyte derived DCs in an FcR1a dependent manner. In addition, removing galactose from ASCA of healthy controls resulted in an increased capacity to activate DCs. Moreover, a DSS colitis model is used to assess the immunostimulatory activity of ASCA like antibodies in mice.

While the clinical data is very interesting, the data on determining the activity of agalactosylated ASCA is preliminary and requires more work.

Major comments:

The notion that agalactosylated ASCA IgG from pre-CD patients modulates monocyte derived DC activity in vitro is interesting but lacks a clinical correlate in patients and thus remains indirect. Is there any evidence that in the peripheral blood of pre-CD patients this type of activation occurs as well, e.g. on NK cells or monocytes in the blood?

Performing a more unbiased experimental approach via incubating for example PBL with different ASCA preparations and doing a scRNAseq or RNAseq of isolated immune cell subsets would be a valuable addition along these lines.

A general (technical) concern I have is that it is not clear if the different forms of ASCA (galactosylated/agalactosylated/active patients/pre-CD patients) have a different tendency to aggregate before and after purification. This might also explain a higher level of activity (FcγR binding).

Along these lines it should be assessed if these different IgG preparations have a different capacity to bind human FcRs, for example via binding to cell lines transfected with individual human FcRs. Further along these lines the ability to activate the complement cascade should be assessed as galactosylation was reported to impact complement pathway activation.

The animal model system (DSS colitis) presented in Fig. 4 is a good start to gain some in vivo insights into the activity of ASCA, however, it remains to be underexplored. Are FcRs or DC play a role here at all? This could be assessed by using FcR deficient mice in which the effect should be diminished. In the DSS model a massive breach of intestinal barriers triggers all sorts of immune activation that could play a role here. Further along these lines: Is the ASCA IgG isolated from immunized mice agalactosylated? Which IgG subclasses are raised? For example IgG3 in the mouse would not bind to FcRs.

A major increase in impact could be achieved by investigating the immunostimulatory activity of ASCA glycoforms in hematopoietic stem cell humanized mice.

The authors mention that they measured an ASCA titer, which seems not to be the case. A titer would be to use different serum/IgG preparation dilutions to determine ASCA abundance. This indeed would be easy to do and helpful to really quantify the response.

Reviewer #2:

Remarks to the Author:

A unique serum glycosylation signature of IgGs predicts the development of Crohn's disease and is associated with pathogenic anti-mannose glycan (ASCA) antibodies. Gaifem et al.,

The authors provide a very interesting body work, investigating homeostatic changes in the serum of CD patients prior to diagnosis. They demonstrate that alterations in the glycome profile of serum IgG Fc portions precede the diagnosis in CD patients and associated with complicated manifestations of CD. They then demonstrate that the altered FC glyco profile pre-disease is able to stimulate pro-inflammatory cytokine production by DCs via FC receptor binding and that removal of glycosylations on ASCA IgG Fc diminishes this DC activation. Similarly NK cell activation through FC receptors is observed resulting high capacity for cytotoxicity and inflammation.

They then develop a mouse model to induce ASCA-IgG and demonstrate that serum transfer of mannan-enriched IgG from immunized mice to healthy controls exacerbates DSS colitis.

The manuscript is very interesting and addresses an understudied aspect in the understanding of CD pathogenesis. The role of ASCA IgG and their mode of action is poorly understood and this study sheds light onto a potential mode of action that can contribute to the development CDs.

The manuscript is well written and the data clearly presented. There are however several gaps and open questions that would strengthen the authors conclusions and improve the provided work. These should all be addressable and not require collection of human samples.

1. It is intriguing that the authors only state changes in IgG glysoylations.
Do the glycosylations on the other isotypes remain unchanged?

2. While the observation is certainly new, the question of why these changes occur remain open.
How can the authors explain the "selective" glycovariants only on IgG?
Do IgG producing plasma cells respond to "subclinical" inflammation prior to diseases while the other plasma cells do not?

3. ASCA antibodies are an indicator of barrier breach. Does a leaky gut modify the IgG-glycosylation?

4. A technical aspect that could require some refinement in the manuscript is the activation of NK cells and DCs. While mannan coating of plates appears reasonable, would other microbial patterns (e.g. bacterial cell wall components), or even total yeast extracts result in a similar activation of NK cells and DCs?

5. The animal model to induce ASCA IgG is a beautiful addition to the manuscript. While the DSS model is interesting, the activation of NK cells and DCs in the gut is lacking leaving a gap between their findings in humans and mice. Analysis of NK cells and DCs should be considered in this model.

Along this line, are murine anti-ASCA IgG antibodies also differentially glycosylated in mannan immunized mice? Would glycosylation changes on murine ASCA-IgG require inflammation or barrier breach as induced by their DSS model?

Do titers of injected ASCA-IgG in their mouse model reflect those seen in patients? Can the exacerbated disease be driven by elevated IgG titers? Similarly to their observations with human serum, could removal of FC glycosylations on mouse ASCA IgGs blunt the effects observed in their DSS model? Alternatively, demonstrating loss of murine ASCA-IgG antibodies through KO lines would be helpful to better connect their findings in humans to their animal model.

6. Lastly, would oral immunization with mannan trigger a similar response? One would assume that intestinal exposure to microbes drives ASCA formation rather than subcutaneous injections?

Author Rebuttal letter:

Dear Dr. Stephanie Houston
Senior Editor Nature Immunology
28th March 2024

We are submitting the revised version R1 of the manuscript NI-A36100-T (now NI-A36100A) entitled:

A UNIQUE SERUM GLYCOSYLATION SIGNATURE OF IGG PREDICTS THE DEVELOPMENT OF CROHN'S DISEASE AND IS ASSOCIATED WITH PATHOGENIC ANTI-MANNOSE GLYCAN (ASCA) ANTIBODIES

The reviewers have made insightful comments that we have addressed and incorporated into the manuscript. All of the revised parts of the manuscript are indicated in red font in the text. In addition, new figures were added and other reformulated, accordingly with the reviewers's suggestions.

We have included Bruno Cavadas as a co-author given his contributions for the bioinformatic analysis of the RNA sequencing assay, as well as Catarina Moreira-Barbosa and Joana Revê's, that were instrumental for patient selection and sample collection, as suggested by the reviewers.

We provide below a point-by-point response to the reviewers's concerns.

We thank you for your consideration on our revised manuscript and hope that this version is now acceptable for publication in Nature Immunology.

We look forward to receiving your opinion.

Sincerely yours,

Salomã S. Pinho, DVM, PhD
Principal Investigator, Institute of Molecular Pathology and Immunology of Univ. Porto (IPATIMUP) & Institute for Research and Innovation in Health (i3S), Univ. Porto, Portugal.
Professor, Medical Faculty, University of Porto, Portugal.
Point-by-point responses to the reviewers's concerns:

Reviewer #1: The manuscript by Gaifem and colleagues addresses a timely and clinically very interesting topic. Understanding the very early phases of autoimmune diseases is critical to allow an early intervention. Ultimately this early intervention may allow resetting the immune system and cure autoimmunity. However, this requires that specific biomarkers predicting disease development become available to justify therapeutic intervention. Gaifem and colleagues study changes in IgG glycosylation in a cohort of patients with Crohn's disease (CD). In CD and more general in inflammatory bowel disease anti-saccharomyces cerevisiae antibodies (ASCA) represent a biomarker for CD. Moreover, changes in serum protein glycosylation and complement pathway activation have been associated with early CD development. The authors make use of a cohort of patient samples collected up to six years before disease development. Interestingly, an agalactoylated IgG glycoform (IgG-G0F) specifically appeared in CD patients but not in patients with UC years before disease development. More importantly, on the IgG2 background this glycovariant correlated with more severe disease development and ASCA IgG. The authors further demonstrate that ASCA isolated from patients before development of disease are able to activate monocyte derived DCs in an FcR1a dependent manner. In addition, removing galactose from ASCA of healthy controls resulted in an increased capacity to activate DCs. Moreover, a DSS colitis model is used to assess the immunostimulatory activity of ASCA like antibodies in mice. While the clinical data is very interesting, the data on determining the activity of agalactosylated ASCA is preliminary and requires more work.

Comment 1.1: The notion that agalactosylated ASCA IgG from pre-CD patients modulates monocyte derived DC activity in vitro is interesting but lacks a clinical correlate in patients and thus remains indirect.

Is there any evidence that in the peripheral blood of pre-CD patients this type of activation occurs as well, e.g. on NK cells or monocytes in the blood?

RESPONSE: We thank the reviewer for raising this question. Our study describes, in a pioneer way, the ability of circulating IgGs (ASCA), generated years before CD diagnosis (and before revealing any symptoms) to imprint a pro-inflammatory response through activation of dendritic cells (DCs) and NK cells. This strategy was implemented not only to mechanistically study ASCA IgG-immune cell interactions, but also because, from PREDICTS cohort, there is no access to peripheral blood samples at the different pre-clinical timepoints.

Nevertheless, to tackle this question raised by the reviewer, we took advantage of SomaLogics data generated in the PREDICTS cohort (Torres & Petralia et al, Gastroenterology 2020) to explore whether activation of these immune cell populations was found in pre-clinical stages. This analysis took into consideration the expression of several markers/proteins to putatively infer about the immune populations of interest. As prior knowledge on cell-type markers, we leveraged the LM22 signature matrix including the expression of different genes for 22 cell types (Newman et al, Nat Methods 2015). Leveraging cell-type signatures from this publication, we computed single-sample gene set enrichment analysis (ssGSEA) via the R package GSVA (Hänzelmann et al, BMC Bioinformatics 2013) (method="zscore") leveraging the proteomic data available for different PREDICT samples. We then correlated the estimated cell-type score with CD/HC status via Wilcoxon Rank-Sum test.

The figure below shows the boxplot of ssGSEA stratified for CD/HC status for different times (i.e., diagnosis, 1-2 years before diagnosis, 2-4 before diagnosis and 6 years before diagnosis). We observe an increase of these cell types (activated DCs and activated NK) in the preclinical phase of CD (p value < 0.05), compared to healthy controls. This increased profile of activated DC and NK cells is significant in all pre-clinical phases but the significance is lost at the closest timepoint from diagnosis, which imply that other inflammatory factors become more prominent at disease diagnosis (see figures below). Interestingly, when we performed the same approach but focusing on monocytes, we observed that there is a significant increase in the signature of these cells, but this only occurs at the timepoints the closest to the diagnosis (figure below). We may thus infer that perhaps the role of monocytes in the inflammatory process is more relevant in a later stage of the pre-clinical phase, and that DCs and NK cells act in earlier stages of the health-to-disease transition process.

These new results were included in Supplementary Figure 5, and in the Results section, line 223, supporting that pre-CD patients exhibit increased activation of DCs and NKs, which is in line with the correlation between an agalactosylated glycoform of ASCA with increased activation of innate immune response, years before CD diagnosis. Methodology was also updated (line 868).

Results:

To gain further insights on the immunological remodeling of DCs, NK cells and B cell-derived plasma cells involved in antibody production occurring in a pre-clinical phase, we analyzed the cell-type signatures from the report of Newman and colleagues 25. We computed single-sample gene set enrichment analysis (ssGSEA) via the R package GSVA 26 leveraging the proteomic data available for different PREDICT samples. We then correlated the estimated cell-type score with CD/HC status. In fact, data from this SomaLogic analysis had shown that markers associated with activated DCs and NK cells were found to be significantly increased in the furthest timepoints from diagnosis in CD patients, compared to other immune populations (Supplementary Fig. 5a,b).

Methods:

Deriving cell-type score from proteomic data. As prior knowledge on cell-type markers, we leveraged the LM22 signature matrix including the expression of different genes for 22 cell types 25. Leveraging cell-type signatures from this study, we computed single-sample gene set enrichment analysis (ssGSEA) via the R package GSVA 26 (method="zscore") using proteomic data available for different PREDICT samples. We then correlated the estimated cell-type score with CD/HC status via Wilcoxon Rank-Sum test.

Comment 1.2: Performing a more unbiased experimental approach via incubating for example PBL with different ASCA preparations and doing a scRNAseq or RNAseq of isolated immune cell subsets would be a valuable addition along these lines.

RESPONSE: We acknowledge the suggestion made by the reviewer concerning this important point. Accordingly with the reviewer's comment, we have performed RNA sequencing of isolated dendritic cells incubated with ASCA IgGs glycoforms. We have chosen DCs as the selected immune cell subset given our mechanistic data (Figure 2) that clearly reveals the biological interaction between ASCA IgGs and DCs immune response. Moreover, as shown for the previous question, activated DCs were pinpointed in the somalogics data as a selected immune signature that is significantly increased in pre-

clinical stages of the disease.

The results demonstrate that DCs co-cultured with ASCA IgGs from 6 years prior to diagnosis (the furthest time point from CD diagnosis) imprint a remarkable gene expression remodeling, with significant upregulation of several signaling pathways, including pro-inflammatory pathways mediated by TNF, NF- κ B, and C-type lectin receptor signaling pathway. These results demonstrate, in an unbiased experimental approach that, in a preclinical phase, immunological alterations are occurring, further supporting the biological relevance of ASCA IgG glycoform in imprinting a pro-inflammatory microenvironment.

We have included the new data on Figure 3, and in the Results section in the manuscript (line 299). We have also updated the Methods section (line 725).

Results:

To further understand the mechanistic impact of ASCA IgGs glycoforms in imposing the reprogramming of immunological pathways in a preclinical phase, we have performed an unbiased experimental approach by doing RNA sequencing analysis of DCs upon co-culture with ASCA IgGs from different time points â 6 years before diagnosis, and at CD diagnosis. Interestingly, we observed a clear differential gene expression profile on DCs imposed by ASCA IgGs at the preclinical phase (6 years before diagnosis) when compared to DCs incubated with ASCA IgGs from HC and from CD at the time of diagnosis (Fig. 3a). Specifically, we found a significant increased expression of genes involved in pro-inflammatory signaling pathways, such as TNF, NF- κ B, and C-type lectin receptor pathways (Fig. 3b), up to 6 years before diagnosis when compared to DCs incubated with ASCA from HC. Indeed, TNF is found to be one of the top genes significantly upregulated by ASCA IgGs in a preclinical phase (Fig. 3c), supporting the premise that ASCA IgGs glycoforms appears to act as an early trigger for health to intestinal inflammation transition.

Methods:

Bulk RNA-seq data processing and analysis

Sequenced reads were aligned to human GRCh38 (ensemble version 94) reference genome using Hisat2 (v2.0.5) with default settings. Counts per gene were generated from the alignment files by FeatureCounts (v1.5.0-p3). Differential expression analysis between pairwise conditions were accessed in R (v4.1.2) using DESeq2 (v1.34). For each comparison, Log2 fold change results were shrunk using the apeglm (v1.23.1) package to remove noise (DE genes with low counts and/or high dispersion values) while preserving significant differences.

Overrepresentation pathway analysis was conducted in clusterProfiler (v4.11.0.001) to the human KEGG ontology database for the set of upregulated and downregulated genes (FDR \leq 0.05), separately. Pathways with an adjusted P value $<$ 0.05, were regarded as significant. Bubble diagrams were constructed for the visualization of the top 6 enriched pathways, after excluding those belonging to human diseases.

Heatmap of the differentially expressed genes between conditions, for the top 6 up-regulated KEGG pathways was plotted using complexHeatmap (v2.10.0). Volcano plots representing differentially expressed genes were generated using ggplot2 (v3.4.0).

Comment 1.3: A general (technical) concern I have is that it is not clear if the different forms of ASCA (galactosylated/agalactosylated/active patients/pre-CD patients) have a different tendency to aggregate before and after purification. This might also explain a higher level of activity (Fc γ R binding).

RESPONSE: We thank the reviewer for this question. We understand the concerns raised by the reviewer since it is described that the low pH normally used in the protocol for protein G isolation may induce the formation of some IgG aggregates that, in turn, may increase the activation via Fc gamma receptors (PMID: 27310175). In fact, previous reports comparing IgG glycosylation profiles, in which two distinct methods of IgG isolation were used (Protein G isolation versus Melon Gel commercially available kit for IgG isolation; PMID: 31681303) has demonstrated that despite there is an increase in IgG aggregation after purification via protein G compared with Melon Gel kits, no alterations in glycosylation profile were found between these two methods of isolation.

To assess this point raised by the reviewer, we measured the aggregation rate in a subset of isolated IgG samples, from the four (pre)clinical groups tested in the functional assays, using a commercially available kit for measuring protein aggregation. As an extra control condition, we have used a purified commercially IgG, aggregated by exposure to higher temperatures. The results showed that our samples exhibit a similar aggregation rate, independently of the experimental group or its glycosylation profile. This supports our rationale that the DC activation observed for ASCA IgGs glycoforms occurs in an antigen-dependent manner, and is not dependent on aggregation of IgGs.

We have included these data in Supplementary Figure 3, as well as in the Methods (line 624) and Results (line 203) sections:

Results:

To exclude any potential effect of IgG aggregation in Fc γ R binding and activity, we measured IgG aggregation upon isolation in the different (pre)clinical conditions. No major differences were observed for all the groups (Supplementary Fig. 3a,b), further supporting that the pro-inflammatory response elicited on DCs upon ASCA IgG glycoform interaction is mediated via Fc γ R interaction.

Methods:

IgG aggregation assay

To assess the degree of IgG aggregation, a subset of IgG samples from healthy controls and CD patients (preclinical phase, at diagnosis, or full-blown disease) were analyzed using a protein aggregation assay kit (Abcam), according to manufacturer's instructions. Besides the standards from the kit, a commercial IgG antibody was incubated at 65°C for 45 minutes to aggregate (positive control). Fluorescence was measured at Ex/Em 440/500 nm and compared with the standard curve of the kit.

Comment 1.4: Along these lines it should be assessed if these different IgG preparations have a different capacity to bind human FcRs, for example via binding to cell lines transfected with individual human FcRs. Further along these lines the ability to activate the complement cascade should be assessed as galactosylation was reported to impact complement pathway activation.

RESPONSE: We thank the reviewer for the comment. Concerning the capacity of these different IgGs preparations to bind to different FcRs, we have selected the Fc-Fc gamma receptor interaction using DCs and NK cells. These immune cell subsets express Fc gamma receptors being particularly involved in IgGs interaction. Fc γ R11a is the main Fc receptor expressed by human NK cells, triggering activation signals that ultimately lead to the mounting of an inflammatory response (PMID: 32719679; PMID: 15030579; PMID: 28813656). For DCs, Fc γ R11a is constitutively expressed and its role in enhanced antigen processing and presentation as well as in pro-inflammatory response is well characterized (PMID: 28813656). Despite not addressing which Fc gamma receptor is the preferential for these IgGs, we have demonstrated that both immune cell types displayed a similar pro-inflammatory effector response upon co-culture with ASCA IgGs from CD patients (when compared with the ones from healthy controls) (Figure 2; Supplementary Fig. 4). Furthermore, we also proved the dependency of ASCA IgG interaction on Fc gamma receptor-mediated signaling by blocking in vitro the Fc γ R11a of DCs using an inhibitory antibody (Fig. 2d,e). These results may set the ground for the future exploitation of the specific binding affinity of ASCA IgGs glycoforms to selected Fc gamma receptors.

Regarding complement activation, this is a very pertinent question raised by the reviewer. In fact, it has been described in the literature that levels of galactosylation on IgG Fc is able to have an impact in complement activation (PMID: 28634480; PMID: 34408013). To tackle this question, we have tested whether ASCA IgGs from CD patients (both preclinical, diagnosis, or established) have increased C1q binding when compared to HC (Supplementary Fig. 3h). In fact, RNAseq data (described in comment #1.2) revealed an association between ASCA IgG from preclinical CD with increased complement activation (C1Q) upon co-culture with DCs, in comparison with HC. This association is in line with the pathogenic properties of ASCA IgG towards health to inflammation transition. In a preclinical phase, we found that ASCA IgG is predominantly agalactosylated (Fig. 1b). To further assess if the glycosylation profile of ASCA IgG is in fact playing a role in this setting of complement modulation, we treated ASCA IgGs isolated from healthy controls with β -galactosidase, to obtain agalactosylated IgGs. Indeed, we observed that the abrogation of galactose residues did not significantly impact the ability of ASCA IgGs to induce complement activation (Supplementary Fig. 3h,i). This result suggests that the impact of ASCA IgG in complement activation that is occurring in a preclinical phase is apparently not mediated by galactose levels of IgG. This association between ASCA IgG and activation of the complement pathway in apparently occurring in a galactose-independent manner, and probably dependent on the antigen binding, an intriguing question that is worth explored in future studies.

We have included this information in Figure 3 and Supplementary Figure 3, as well as in the Results section (line 312). The methodology was also updated (line 742).

Results:

In addition, one of the pathways shown to be upregulated by preclinical ASCA IgGs is the complement cascade (Fig. 3b), particularly C1QA and C1QB (Fig. 3c). This goes in line with our results demonstrating that ASCA IgGs from CD patients (both preclinical, diagnosed and established) exhibit an increased C1q binding (Supplementary Fig. 3h), which is in line with the pro-inflammatory environment imposed by ASCA IgG in pre-CD. This complement activation observed in pre-CD is apparently independent of galactose levels on ASCA IgG (Supplementary Fig. 3i).

Methods:

C1q (complement) binding assay

96-well Nunc $\text{\textcircled{R}}$ MaxiSorp $\text{\textcircled{R}}$ high-binding plates were pre-coated with mannan from *Saccharomyces cerevisiae* (SigmaAldrich, St-Louis, MO, USA) at 10 μ g/mL in 50 mM carbonate $\text{\textcircled{R}}$ carbonate coating buffer (pH 9.6), overnight, at 4°C. Afterwards, a blocking step with 5% of BSA was performed. 6 μ g/mL of total IgGs were added to plates previously coated with mannan (10 μ g/mL), for 90 minutes at 37°C in a humidified incubator with 5% of CO $_2$. After three washing steps with PBS 3% BSA 0.05% Tween20, complement component C1q was added (10 μ g/mL) for 1 hour at room temperature. After three washing steps (as previously), HRP sheep anti-human C1q was added for incubation for 1 hour at room temperature. Plate was washed four times (as previously), and 3,3',5,5'-tetramethylbenzidine (substrate

solution) was added for 15 min. Reaction was stopped with 1 N H₂SO₄ and absorbance was measured at 450 nm.

Comment 1.5: The animal model system (DSS colitis) presented in Fig. 4 is a good start to gain some in vivo insights into the activity of ASCA, however, it remains to be underexplored.

Are FcRs or DC play a role here at all? This could be assessed by using FcR deficient mice in which the effect should be diminished. In the DSS model a massive breach of intestinal barriers triggers all sorts of immune activation that could play a role here.

Further along these lines: Is the ASCA IgG isolated from immunized mice agalactosylated? Which IgG subclasses are raised? For example IgG3 in the mouse would not bind to FcRs.

RESPONSE: We agree with the reviewer's comments. In fact, we have demonstrated that ASCA IgGs from CD patients, both preclinical and at diagnosis, are able to impose an inflammatory response in DCs. We have shown that this effect on DCs is mediated by FcγRIIIa, since the blockade of this receptor using specific inhibitory antibodies abrogates the proinflammatory response of DCs (Figure 2e). To tackle this question raised by the reviewer, we have analyzed the effects of ASCA IgG isolated from immunized mice in bone marrow-derived DCs (BMDCs) activation. The results showed that BMDCs incubated in vitro with murine ASCA IgGs display an increased expression of MHC-II (Fig. 4e), which is also in line with increased susceptibility to intestinal inflammation observed in vivo (Fig. 4b). In addition, and to gain mechanistic insights on the capacity of ASCA IgG in activating inflammatory pathways via FcγR interaction, we have used a full FcγR KO mice that was adoptively transferred with ASCA IgG isolated from mannan immunized mice. Interestingly, the results demonstrated that the susceptibility to colitis in FcγR KO mice inoculated with ASCA was clearly diminished in comparison with WT, with FcγR KO mice displaying a significant decrease in disease severity and disease activity index (DAI) (Figure 4f), alongside with a significant decreased levels of pro-inflammatory IL-1β in the supernatants of colonic explants (Fig. 4g). These results clearly support the biological effect of ASCA IgG in imposing a pro-inflammatory environment associated with intestinal inflammation and the relevance of ASCA IgG Fc-γR interaction.

We have included this topic in the results section (line 401 and 411) and in Figure 4. Methodology was also updated (lines 802 and 826).

Results:

Intestinal inflammatory infiltrate was analyzed in ASCA-treated versus non-treated mice, and the isolation of lamina propria leukocytes demonstrated an increased expression of MHC-II+ DCs (Fig. 4d), suggesting the impact of ASCA IgGs in imposing an activation profile of DCs in vivo. Additionally, co-culture of ex vivo ASCA IgGs from mannan-immunized mice with bone-marrow-derived dendritic cells (BMDCs) from WT mice showed a similar increased expression of MHC-II in DCs (Fig. 4e).

(â)

In addition, and to gain mechanistic insights on the biological effect of ASCA IgG in imposing an inflammatory environment associated with intestinal inflammation through a mechanism dependent on Fc-γR interaction, we used a mouse model deficient for Fc receptor γ (FcγRI/II/III/IV KO; simplified here as FcγR KO). WT and FcγR KO were inoculated with ASCA IgGs isolated from mannan immunized mice (as performed above), and colitis susceptibility upon DSS treatment was evaluated. We demonstrated that FcγR KO mice inoculated with ASCA displayed a significant decreased susceptibility to develop colitis as revealed by decreased DAI in comparison with WT (Fig. 4f; Supplementary Fig. 7g), alongside with a significant decreased levels of IL-1β in the supernatants of colonic explants (Fig. 4g). TNF-α levels are also decrease, despite not reaching statistical significance (Supplementary Fig. 7h). Taken together, these results demonstrate the relevance of ASCA IgG Fc-γR interactions in imposing a pro-inflammatory environment associated with intestinal inflammation and colitis susceptibility, bringing to light the pathogenic effects of ASCA IgGs in vivo.

Methods:

Wild-type, and mice deficient for Fc receptor γ (FcγRI/II/III/IV KO; simplified here as FcγR KO), with 10-16 weeks (recipient) were intraperitoneally injected with 100 μg of ASCA-enriched IgG pool or, as control, 100 μg PBS-derived IgGs, at day 0 and day 3. At day 0, colitis was induced chemically by 2% DSS (36,000-50,000Da; MP Biomedicals) administration on the drinking water ad libitum from day 0 to day 7, and then switched to tap water. Mice were monitored daily and disease activity score was assessed according to parameters in Supplementary Table 5, and were euthanized according to the humane endpoints.

(â)

Bone-marrow derived dendritic cells (BMDCs) and co-culture with murine ASCA IgGs

Bone marrow was extracted from the femur and tibias of C57BL/6 WT mice, and isolated cells were

then cultured in RPMI supplemented with 10% FBS, penicillin/streptomycin (100 U/mL), and mouse recombinant proteins IL-4 and GM-CSF (50 ng/mL; E. coli, PeproTech). On the third day of culture, the medium was replaced, and the cells were in cultured for 4 more days. Differentiated DCs were then incubated with murine ASCA IgGs, and PBS IgGs (control), similarly to the protocol described for IgG-dependent human dendritic cell activation. Cells were acquired by flow cytometry on a Cytex Aurora system and data were analyzed in FlowJo version 10.5.3 (Tree Star, Inc., Ashland, OR).

Regarding murine ASCA IgG glycosylation profile, raised by the reviewer, we have collected ASCA IgGs from mice immunized with mannan and the glycosylation profile was assessed by nanoLC-ESI-MS similarly with the approach used for the human IgG glycome characterization. Samples were analyzed as a pool of several mice sera. The results showed that ASCA IgGs isolated from immunized mice are glycosylated but no major alterations were found in terms of glycans composition of IgGs comparing mannan immunized versus non-immunized mice (total IgG). In any case, changes in serum IgG glycome in immunized mice were rather unexpected, since we were using wild-type mice with no basal genetic alterations in terms of glycosylation machinery on B cells. The changes in serum IgG glycosylation observed in the human setting were detected in the context of a preclinical disease/pro-inflammatory environment, which is not the case in the animal model that we used.

Accordingly with the reviewer's comment, we have included these results in Supplementary Figure 7 and in the Results section (line 389). Methodology was also updated (line 593).

The serum IgG glycome from immunized versus non-immunized mice was assessed and the results showed that no major differences were found in terms of glycan composition of IgGs comparing immunized versus non-immunized mice (Supplementary Fig. 7c).

Methods:

For human IgA, and murine IgGs: after sample loading, the trap column was switched in-line with the gradient and C18 nano LC column (150 mm x 100 µm i.d., 2.6 µm SunShell core shell particles; Chromanik Technologies Inc.) for 9 minutes. Trap column was cleaned with two full loop injections containing 20 µl of 95 % ACN and 50% ACN in IPA respectively. C18 nano-LC column was equilibrated with 100 % mobile phase A (0.1% TFA), for 2 minutes. IgG glycopeptides were reconstituted in 30 µl of ultrapure water before injection. Separation was achieved at 1 ml/min using the following gradient of mobile phase A and mobile phase B (80 % ACN and 20 % 0.1 % TFA respectively): 1 - 1.5 min 0 % B - 18.5 % B, 1.5 - 5 min 18.5 % B - 26 % B, 5 - 9 min 26 % B. The ACQUITY M-Class system was coupled with Compact mass spectrometer (Bruker Daltonics, Bremen, Germany) equipped with Captive Spray ion source and nano Booster for introduction of acetonitrile vapor into the source. Nitrogen was used as a drying gas (4 l/min) and nebulizer (0.2 bars). Quadrupole and collision energies were set to 4 eV. Spectra were recorded from m/z 600 to 2500 with 2 averaged scans at a frequency of 0.5 Hz. Acquity M-class UPLC system was operated under MassLynx software 4.1 while Bruker Compact Q-TOF-MS was operated under HyStar software version 4.1. The detected glycopeptides are presented in Supplementary Table 2, and in accordance with previous reports^{47, 48, 49}.

Comment 1.6: A major increase in impact could be achieved by investigating the immunostimulatory activity of ASCA glycoforms in hematopoietic stem cell humanized mice.

RESPONSE: We acknowledge the importance of the reviewer's perspective on validating the mechanism in a humanized mouse. Despite interesting and worth-explored in future studies, this is a very demanding and risk approach. We believe that, at this stage, the compelling body of experimental data generated in this study, combining both in vitro, ex vivo, in vivo approaches further validated in a precious cohort from pre-CD patients - the PREDICTS cohort - allow us to propose a new predictive biomarker that underlies a new pro-inflammatory mechanism driven by ASCA IgG glycome and innate immune response, taking place many years before CD diagnosis. In addition, the new data generated using the FcγR deficient mice allowed us to gain further mechanistic insights on the immunostimulatory activity and pathogenic role of ASCA IgG in IBD development.

Comment 1.7: The authors mention that they measured an ASCA titer, which seems not to be the case. A titer would be to use different serum/IgG preparation dilutions to determine ASCA abundance. This indeed would be easy to do and helpful to really quantify the response.

RESPONSE: We agree with the reviewer's comment. As suggested, we have used different serum/IgG preparation dilutions to determine ASCA abundance. The information displayed in new supplementary Figure 1b is related with ASCA relative quantification (arbitrary units) and therefore we have changed the y axis legend to "ASCA levels". To complement this information, we have assessed ASCA titers for a subset of samples.

This data is now embedded as Supplementary Figure 1, and in the results section (line 188). Methodology was also updated (line 541).

Results:

The same profile is found for the titers of ASCA, specifically analyzed for a subset of samples, in which a clear increase was found from HC to preclinical and CD samples (Supplementary Fig. 1d).

Methods:

ASCA levels and titers (serial dilutions) were measured in human serum samples using QUANTA Lite ASCA IgG ELISA (QUANTA), according to manufacturer's instructions.

Reviewer #2: A unique serum glycosylation signature of IgGs predicts the development of Crohn's disease and is associated with pathogenic anti-mannose glycan (ASCA) antibodies.

The authors provide a very interesting body work, investigating homeostatic changes in the serum of CD patients prior to diagnosis. They demonstrate that alterations in the glycome profile of serum IgG Fc portions precede the diagnosis in CD patients and associated with complicated manifestations of CD. They then demonstrate that the altered FC glyco profile pre-disease is able to stimulate pro-inflammatory cytokine production by DCs via FC receptor binding and that removal of glycosylations on ASCA IgG Fc diminishes this DC activation. Similarly NK cell activation through FC receptors is observed resulting high capacity for cytotoxicity and inflammation.

They then develop a mouse model to induce ASCA-IgG and demonstrate that serum transfer of mannan-enriched IgG from immunized mice to healthy controls exacerbates DSS colitis.

The manuscript is very interesting and addresses an understudied aspect in the understanding of CD pathogenesis. The role of ASCA IgG and their mode of action is poorly understood and this study sheds light onto a potential mode of action that can contribute to the development CDs.

The manuscript is well written and the data clearly presented. There are however several gaps and open questions that would strengthen the authors conclusions and improve the provided work. These should all be addressable and not require collection of human samples.

Comment 2.1: It is intriguing that the authors only state changes in IgG glycosylations. Do the glycosylations on the other isotypes remain unchanged?

RESPONSE: We acknowledge the reviewer's question. To tackle this point, sera from a subset of pre-CD patients at different time-points before disease onset and at diagnosis (n=8/group) was analyzed regarding IgA glycome composition. IgA was selected as another immunoglobulin isotype given the prominent role of IgA in gut mucosa immunity. The results showed the existence of a differential serum IgA glycome comparing pre-CD and HC, however alterations in agalactosylated glycoforms in IgA was only detected at diagnosis and not in pre-disease state (-6 years before diagnosis) as consistently observed in IgG (Figure 1). Specifically, the agalactosylated signature that was clearly changed in IgG (specifically the H3N4F1) was not detected in IgA, both at preclinical or at diagnosis timepoints. Indeed, significant alterations in glycosylation profile of IgA are rather found at diagnosis, which suggests that the glycome signature described to be associated with preclinical CD is specific for IgG. Nevertheless, and to further validate this observation, the analysis of IgA glycosylation in a larger subset of samples would be needed in future studies aiming to analyze the relevance of IgA glycosylation in IBD, a topic that is out of the scope of this manuscript.

Accordingly with the reviewer's suggestion, we have included this data on Supplementary Figure 1, as well as in the Results section (line 158). Methodology was also updated (line 593).

Results:

This agalactosylated profile seems to be specific for serum IgG, since no major alterations were found for agalactosylated forms of serum IgA at the preclinical stage (Supplementary Fig. 1b).

Methods:

For human IgA, and murine IgGs: after sample loading, the trap column was switched in-line with the gradient and C18 nano LC column (150 mm x 100 μ m i.d., 2.6 μ m SunShell core shell particles; Chromanik Technologies Inc.) for 9 minutes. Trap column was cleaned with two full loop injections containing 20 μ l of 95 % ACN and 50% ACN in IPA respectively. C18 nano-LC column was equilibrated with 100 % mobile phase A (0.1% TFA), for 2 minutes. IgG glycopeptides were reconstituted in 30 μ l of ultrapure water before injection. Separation was achieved at 1 ml/min using the following gradient of mobile phase A and mobile phase B (80 % ACN and 20 % 0.1 % TFA respectively): 1 \rightarrow 1.5 min 0 % B

18.5 % B, 1.5 - 5 min 18.5 % B to 26 % B, 5 - 9 min 26 % B. The ACQUITY M-Class system was coupled with Compact mass spectrometer (Bruker Daltonics, Bremen, Germany) equipped with Captive Spray ion source and nano Booster for introduction of acetonitrile vapor into the source. Nitrogen was used as a drying gas (4 l/min) and nebulizer (0.2 bars). Quadrupole and collision energies were set to 4 eV. Spectra were recorded from m/z 600 to 2500 with 2 averaged scans at a frequency of 0.5 Hz. Acquity M-class UPLC system was operated under MassLynx software 4.1 while Bruker Compact Q-TOF-MS was operated under HyStar software version 4.1. The detected glycopeptides are presented in Supplementary Table 2, and in accordance with previous reports^{47, 48, 49}.

Comment 2.2: While the observation is certainly new, the question of why these changes occur remain open.

How can the authors explain the selective glycovariants only on IgG?

Do IgG producing plasma cells respond to subclinical inflammation prior to diseases while the other plasma cells do not?

RESPONSE: We acknowledge the reviewer's question. Regarding the selective glycovariants in IgG, we cannot exclude that other immunoglobulin isotypes may be also affected by changes in glycosylation such as in IgA, as described above. However, the role of different Ig glycovariants should be study at a specific level as it may be different accordingly with the physio-pathological context. In fact, while our understanding of the regulation of protein glycosylation is limited, recent studies demonstrated that gene networks that regulate glycosylation of different proteins can be very different (Landini et al, Nat Commun 2022). Large study on over 12,000 individuals identified 27 genetic loci that associate with IgG glycosylation, including some that are known risk factors for IBD (Klaric et al, Sci Advances 2020). Therefore, specific glycovariants on IgG that associate with IBD development are thus expected, as demonstrated in this study. To tackle this important question raised by the reviewer, a statement in the discussion section was included (line 470):

In fact, while our understanding of the regulation of protein glycosylation is limited, recent studies demonstrated that gene networks that regulate glycosylation of different proteins can be very different⁴². Large study on over 12,000 individuals identified 27 genetic loci that associate with IgG glycosylation, including some that are known risk factors for IBD⁴³. Therefore, specific glycovariants on IgG that associate with IBD development are revealed in this study.

Regarding the question on IgG-producing plasma cells dynamics in a subclinical inflammation, we tackled this important question raised by the reviewer in two complementary ways. First, we resort to our data from somalogics markers at pre-CD stage (from PREDICTS cohort), and the results demonstrate an increase in plasma cells in preclinical phase, significantly observed in the 2/4-year timeframe before CD diagnosis. This may suggest that at preclinical stages, there is already a potential remodeling of the immune cell populations, including plasma cells, within the context of a subclinical inflammation, that may contribute to trigger intestinal inflammation. The data on plasma cells is now included in supplementary figure 3c. To further complement this information, and given the fact that blood or PBMCs are not available from military individuals from PREDICTS cohort, we analyzed PBMCs derived from patients with inaugural disease (newly diagnosed CD, mimicking health to inflammation transition), as well as from first-degree relatives (FDR) of CD patients, focusing on B cells, plasma cells and plasmablasts. These B cell populations are widely described as contributing for IBD pathogenesis and antibodies production (PMID: 35190725; PMID: 24835396). Interestingly, the results showed that there is a significant increase in the frequency of B cells and plasmablasts from FDR to CD patients, concomitantly with a trend increase in the frequency of plasma cells, which support that in health to intestinal inflammation transition there is the remodeling of B cell compartment.

We have added these new data in Supplementary Figure 5, in the results section (line 232), in the methodology (line 526, 701 and 868), and in a new table referring the patients's demographic data (Supplementary Table 4):

Results:

In addition, we also found that markers for plasma cells were significantly increased in the 2/4-year timeframe before diagnosis, supporting the existence of a potential remodeling of the B cell-derived plasma cells, within the context of a subclinical inflammation, that may contribute to autoantibody production associated with intestinal inflammation (Supplementary Fig. 5c). To complement this observation, we analyzed the frequency of B cells, plasma cells and plasmablasts in PBMCs isolated from blood of a subset of newly diagnosed CD patients, as well as in first-degree relatives (FDR), a risk population for IBD development and a proxy of a health to intestinal transition setting (Supplementary Table 4). Interestingly, we observed a significant increase in the frequency of B cells and plasmablasts from FDR to CD patients, concomitantly with a trend increase in the frequency of plasma cells. No major alterations were found in IgG-producing plasma cells in these individuals (Supplementary Fig. 5d-g).

Methods:

For the analysis of plasma cells, peripheral blood mononuclear cells (PBMCs) from CD patients (collected at the diagnosis of inaugural disease), first-degree relatives of CD patients, and healthy controls were obtained both at Centro Hospitalar Universitário Santo António, Porto, and Hospital Beatriz Ângelo, Loures (Supplementary Table 4).

(â)

Plasma cell characterization in human PBMCs

PBMCs from inaugural CD patients (collected at the time of diagnosis), first-degree relatives of CD patients, and healthy controls were collected and the plasma cell characterization was performed by flow cytometry, using surface and intracellular markers (Supplementary Table 6), as described above. Cells were acquired by flow cytometry on a Cytex Aurora system and data were analyzed in FlowJo version 10.5.3 (Tree Star, Inc., Ashland, OR), using the gating strategy shown in Supplementary Fig. 9c.

(â)

Deriving cell-type score from proteomic data. As prior knowledge on cell-type markers, we leveraged the LM22 signature matrix including the expression of different genes for 22 cell types. Leveraging cell-type signatures from this study, we computed single-sample gene set enrichment analysis (ssGSEA) via the R package GSVA26 (method="zscore") using proteomic data available for different PREDICT samples. We then correlated the estimated cell-type score with CD/HC status via Wilcoxon Rank-Sum test.

Comment 2.3: ASCA antibodies are an indicator of barrier breach. Does a leaky gut modify the IgG-glycosylation?

RESPONSE: We thank the reviewer for raising this pertinent point. To address this question, we have assessed the integrity of the intestinal epithelial barrier (by FITC-dextran assay) of immunized mice with mannan. Despite significant increased levels of serum ASCA IgG produced upon mannan immunization, there were no significant alterations in intestinal permeability in these mice (Supl. Fig.7).

Regarding the impact in murine IgG glycosylation profile (a similar question was raised by Reviewer#1 - see comment 1.4), we have collected ASCA IgGs from mice immunized with mannan and the glycosylation profile was assessed by nanoLC-ESI-MS similarly with the approach used for the human IgG glycome characterization. Samples were analysed as a pool of several mice sera. The results showed that no major alterations were found in terms of glycans composition of IgGs comparing mannan immunized versus non-immunized mice (total IgG).

In both cases, changes in intestinal permeability and/or serum IgG glycome in immunized mice were rather unexpected, since we were using wild-type mice with no basal alterations in mucosa barrier and/or genetic alterations in terms of glycosylation machinery on B cells. The association between ASCA and the impact in intestinal permeability (in the human IBD setting) as well as the changes in serum IgG glycosylation observed in the human setting were detected in the context of a preclinical disease/pro-inflammatory environment, which is not the case in the animal model that we used for ASCA production.

Accordingly with the reviewer's suggestion, this data is displayed in Supplementary Figure 7 and in the Results section (line 382). Methods were also updated (line 593 and 796).

Results:

WT mice were immunized with mannan (subcutaneous inoculation), in order to stimulate the production of anti-mannan (ASCA-like) IgGs. Mannan immunization resulted in significant increased levels of serum ASCA IgGs (Supplementary Fig. 7a). No significant alterations in intestinal permeability upon mannan immunization were observed (Supplementary Fig. 7b). Serum was collected from immunized mice and ASCA-enriched IgGs were isolated (alongside with the PBS-immunized IgGs from the control group). The serum IgG glycome from immunized versus non-immunized mice was assessed and the results showed that no major differences were found in terms of glycan composition of IgGs comparing immunized versus non-immunized mice (Supplementary Fig. 7c).

Methods:

For human IgA, and murine IgGs: after sample loading, the trap column was switched in-line with the gradient and C18 nano LC column (150 mm x 100 µm i.d., 2.6 µm SunShell core shell particles; Chromanik Technologies Inc.) for 9 minutes. Trap column was cleaned with two full loop injections containing 20 µl of 95 % ACN and 50% ACN in IPA respectively. C18 nano-LC column was equilibrated with 100 % mobile phase A (0.1% TFA), for 2 minutes. IgG glycopeptides were reconstituted in 30 µl of ultrapure water before injection. Separation was achieved at 1 ml/min using the following gradient of mobile phase A and mobile phase B (80 % ACN and 20 % 0.1 % TFA respectively): 1 to 1.5 min 0 % B to 18.5 % B, 1.5 - 5 min 18.5 % B to 26 % B, 5 - 9 min 26 % B. The ACQUITY M-Class system was

coupled with Compact mass spectrometer (Bruker Daltonics, Bremen, Germany) equipped with Captive Spray ion source and nano Booster for introduction of acetonitrile vapor into the source. Nitrogen was used as a drying gas (4 l/min) and nebulizer (0.2 bars). Quadrupole and collision energies were set to 4 eV. Spectra were recorded from m/z 600 to 2500 with 2 averaged scans at a frequency of 0.5 Hz. Acquity M-class UPLC system was operated under MassLynx software 4.1 while Bruker Compact Q-TOF-MS was operated under HyStar software version 4.1. The detected glycopeptides are presented in Supplementary Table 2, and in accordance with previous reports^{47, 48, 49}.

(â!)

In vivo intestinal permeability was assessed by administration of fluorescein isothiocyanate (FITC)-labelled dextran. Food and water were withdrawn for 8 hours. Mice were administered 44 mg/100 g body weight of FITC-labelled dextran (TdB Consultancy; 4 kDa) by oral gavage. Serum was collected, from the tail vein, four hours later and fluorescence intensity was measured by spectrophotofluorimetry (excitation: 485 nm; emission: 528 nm).

Comment 2.4: A technical aspect that could require some refinement in the manuscript is the activation of NK cells and DCs. While mannan coating of plates appears reasonable, would other microbial patterns (e.g. bacterial cell wall components), or even total yeast extracts result in a similar activation of NK cells and DCs?

RESPONSE: We agree with the reviewer in testing other microbial patterns. To answer this question, we have followed a similar approach as for the mannan coating, but using other microbial-derived component such as the yeast/bacteria-derived Î²-glucan (Î²-D-glucose polysaccharides). DC activation was analyzed as described for mannan coatings. We have observed that contrary to mannan-coating settings (for ASCA selection), Î²-glucan-specific IgGs did not impose a differential pro-inflammatory/activated profile on DCs (Supplementary Figure 6d). This actually goes in line with the data shown in the manuscript, in which we have tested another glycan-antigen, di-GlcNAc glycodendrimer (Figure 2o; Supplementary Figure 6b), in which we actually observe a decrease in DC activation with IgGs from CD patients.

Similar profile is found for NK cells (Supplementary Fig. 4f,g). Co-cultures of NK cells with Î²-glucan-specific IgGs from CD patients, resulted in a decreased NK cell activation through decrease in CD107a expression. Accordingly, NK cells co-cultured with IgGs specific for di-GlcNAc glycodendrimer showed no alterations in terms of activation. Collectively, these results suggest that mannan recognition is selectively imposing ASCA-specific activation of DCs and NK cells.

This result was included in Supplementary Figures 4 and 6 and in the Results section (line 281). The current methodology was also updated (lines 671 and 688).

Results:

Furthermore, and to validate the specific effect of anti-mannose (ASCA) IgG in activating DCs, we tested incubation of IgGs with other microbial patterns, in particular the yeast/bacteria-derived Î²-glucan. We demonstrated that anti-Î²-glucan antibodies showed no differential activation of DCs upon co-culture (Supplementary Fig. 6d). The selective activation imposed by ASCA IgGs was also observed for NK cells, since no major alterations were found for NK cells co-cultured with di-GlcNAc-specific IgGs or with Î²-glucan-specific IgGs (Supplementary Fig. 4f,g).

Supplementary Fig. 6 (DCs):

Supplementary Fig. 4 (NK cells):

Methods:

6 Åµg/mL of total IgGs were added to plates previously coated with mannan, Î²-glucan or di-GlcNAc (10 Åµg/mL), for 90 minutes at 37Å°C in a humidified incubator with 5% of CO₂.

(â!)

Purified NK cells were suspended in complete RPMI medium supplemented with IL-2 (100 U/mL, Pepotrech) and plated in wells previously coated with mannan, Î²-glucan or di-GlcNAc (as described above for DCs), at 1x10⁵ cells/well in the presence of anti-human CD107a (LAMP-1; Supplementary Table 6).

Comment 2.5: The animal model to induce ASCA IgG is a beautiful addition to the manuscript. While the DSS model is interesting, the activation of NK cells and DCs in the gut is lacking leaving a gap between their findings in humans and mice. Analysis of NK cells and DCs should be considered in this model.

Along this line, are murine anti-ASCA IgG antibodies also differentially glycosylated in mannan immunized mice? Would glycosylation changes on murine ASCA-IgG require inflammation or barrier breach as induced by their DSS model?

RESPONSE: We thank the reviewer for highlighting the in vivo approach used in this study. Accordingly

with the ReviewerA's suggestion, we have analyzed DCs and NK cells in mice inoculated with ASCA IgGs. We demonstrated a significant increased expression of MHC-II⁺ dendritic cells in the colonic mucosa of ASCA-inoculated mice in comparison with control, which suggests that there is an activation of DCs upon ASCA administration in vivo. The ability of murine ASCA to activate DCs was further validated in vitro, by co-culture of ASCA IgG with bone-marrow derived DCs from mice, and the results showed a similar activation profile (Figure 4). For NK cells, there is also a similar trend of activation of these cells imposed by ASCA IgGs, through a significant increased expression of Granzyme B, with IFN γ -producing NK cells being also increased however not raising statistical significance (Supplementary Fig. 7 e,f).

The data was included in Figure 4, Supplementary Figure 7 and in the Results section (line 401). Methods were updated on line 810.

Results:

Intestinal inflammatory infiltrate was analyzed in ASCA-treated versus non-treated mice, and the isolation of lamina propria leukocytes demonstrated an increased expression of MHC-II⁺ DCs (Fig. 4d), suggesting the impact of ASCA IgGs in imposing an activation profile of DCs in vivo. Additionally, co-culture of ex vivo ASCA IgGs from mannan-immunized mice with bone-marrow-derived dendritic cells (BMDCs) from WT mice showed a similar increased expression of MHC-II in DCs (Fig. 4e). For NK cells, there is a similar trend activation imposed by ASCA IgGs, through a significant increased expression of Granzyme B (Supplementary Fig. 7e). The frequency of IFN γ -producing NK cells is also increased, however not raising statistical significance (Supplementary Fig. 7f).

Figure 4:

Supplementary Figure 7:

Methods:

For the isolation of lamina propria leukocytes, colonic fragments of 0.5-1 cm were incubated in DMEM medium supplemented with 1mM CaCl₂, 1mM MgCl₂, 1.5 mg/mL of collagenase IV (Sigma), and 0.4 mg/mL of dispase (Gibco), under 100 rpm agitation at 37°C for 40 minutes. Tissues were dissociated and filtered through a 70 μ m cell strainer (BD Biosciences). Cell suspension was resuspended in RPMI 1640 medium supplemented with 10% FBS, and 1% penicillin/streptomycin, and layered upon Lymphoprep solution in a proportion of 1:2 (Lymphoprep:cell suspension). After gradient centrifugation at 800g for 20 minutes at 20°C (without acceleration or break), immune cells (retained in the interface) were collected for the staining using the antibodies described in Supplementary Table 6. Cells were acquired by flow cytometry on a Cytex Aurora system and data were analyzed in FlowJo version 10.5.3 (Tree Star, Inc., Ashland, OR), using the gating strategy shown in Supplementary Fig. 9d.

Regarding the question on whether murine anti-ASCA IgG antibodies are differentially glycosylated in mannan immunized mice, and if for that a barrier breach is required, this question is answered above in comment 2.3.

As mentioned for the 2.2 question, this data is displayed in Supplementary Figure 7 and in the Results section (line 383), as well as in the Methods section (lines 593 and 796).

Results:

WT mice were immunized with mannan (subcutaneous inoculation), in order to stimulate the production of anti-mannan (ASCA-like) IgGs. Mannan immunization resulted in significant increased levels of serum ASCA IgGs (Supplementary Fig. 7a). No significant alterations in intestinal permeability upon mannan immunization were observed (Supplementary Fig. 7b). Serum was collected from immunized mice and ASCA-enriched IgGs were isolated (alongside with the PBS-immunized IgGs from the control group). The serum IgG glycome from immunized versus non-immunized mice was assessed and the results showed that no major differences were found in terms of glycan composition of IgGs comparing immunized versus non-immunized mice (Supplementary Fig. 7c).

Methods:

For human IgA, and murine IgGs: after sample loading, the trap column was switched in-line with the gradient and C18 nano LC column (150 mm x 100 μ m i.d., 2.6 μ m SunShell core shell particles;

Chromanik Technologies Inc.) for 9 minutes. Trap column was cleaned with two full loop injections containing 20 μ l of 95 % ACN and 50% ACN in IPA respectively. C18 nano-LC column was equilibrated with 100 % mobile phase A (0.1% TFA), for 2 minutes. IgG glycopeptides were reconstituted in 30 μ l of ultrapure water before injection. Separation was achieved at 1 ml/min using the following gradient of mobile phase A and mobile phase B (80 % ACN and 20 % 0.1 % TFA respectively): 1 \rightarrow 1.5 min 0 % B \rightarrow 18.5 % B, 1.5 - 5 min 18.5 % B \rightarrow 26 % B, 5 - 9 min 26 % B. The ACQUITY M-Class system was coupled with Compact mass spectrometer (Bruker Daltonics, Bremen, Germany) equipped with Captive Spray ion source and nano Booster for introduction of acetonitrile vapor into the source. Nitrogen was used as a drying gas (4 l/min) and nebulizer (0.2 bars). Quadrupole and collision energies were set to 4 eV. Spectra were recorded from m/z 600 to 2500 with 2 averaged scans at a frequency of 0.5 Hz. Acquity M-class UPLC system was operated under MassLynx software 4.1 while Bruker Compact Q-TOF-MS was operated under HyStar software version 4.1. The detected glycopeptides are presented in Supplementary Table 2, and in accordance with previous reports^{47, 48, 49}.

(a)

In vivo intestinal permeability was assessed by administration of fluorescein isothiocyanate (FITC)-labelled dextran. Food and water were withdrawn for 8 hours. Mice were administered 44 mg/100 g body weight of FITC-labelled dextran (TdB Consultancy; 4 kDa) by oral gavage. Serum was collected, from the tail vein, four hours later and fluorescence intensity was measured by spectrophotofluorimetry (excitation: 485 nm; emission: 528 nm).

Comment 2.5: Do titers of injected ASCA-IgG in their mouse model reflect those seen in patients? Can the exacerbated disease be driven by elevated IgG titers? Similarly to their observations with human serum, could removal of Fc glycosylations on mouse ASCA IgGs blunt the effects observed in their DSS model? Alternatively, demonstrating loss of murine ASCA-IgG antibodies through KO lines would be helpful to better connect their findings in humans to their animal model.

RESPONSE: We thank the reviewer for this question. There is a wide range of IgG concentrations described for in vivo inoculation in mice (PMID: 19664634; PMID: 32161277). The levels of administered IgGs were taken into consideration previous reports on IgG inoculation in mice models (PMID: 35235456; PMID: 16912303).

Regarding the comment about removal of Fc glycosylation of murine ASCA IgGs followed by their in vivo administration, despite scientifically sound, it would impose a major technical limitation since the selective removal or remodeling of Fc glycans (through e.g. β -galactosidase S) without perturbing Fab glycosylation of IgG is not possible. Fab glycosylation is essential for stability, half-life, and binding characteristics of IgGs to the antigen. Therefore, the ex vivo remodeling of IgG glycosylation would not be specific of Fc domain, imposing major alterations in stability, specificity and half-life of IgG due to changes in Fab domain. Moreover, the amount of murine IgGs needed for the ex vivo glycosylation remodeling (a tremendous amount of enzyme would be needed for depletion of galactose residues) followed by in vivo administration limits the technical execution of this approach.

Likewise, the selective depletion of ASCA-IgG antibodies through KO lines, suggested by the reviewer is technically very challenging and demanding. ASCA is an anti-glycan antibody, in which B cell producing antibodies recognize mannose-enriched epitopes that can be present in host cells or in a variety of different microorganisms. The specific depletion of B cells producing anti-mannose antibodies (ASCA) would impose major technical and experimental efforts that, despite risky would be worth explored in another study.

Nevertheless, the set of new results generated in this revised version through unbiased RNAseq data in human cells and the use of Fc γ R KO mice among many others, further validate and strongly support the main findings of this study on the pathogenic role of ASCA IgG in health to inflammation transition and CD development.

Comment 2.6: Lastly, would oral immunization with mannan trigger a similar response? One would assume that intestinal exposure to microbes drives ASCA formation rather than subcutaneous injections?

RESPONSE: We thank the reviewer for the suggestion to test oral immunization with mannan. The impact of mannan via oral supplementation is an interesting approach to mimic the intestinal exposure of mannan-derived epitopes. Oral immunization was performed with mannan administration via oral gavage in a 3-day scheme administration, repeated during 3 weeks (Supplementary Fig. 8a) (based also in literature such as PMID: 30806148 and PMID: 35460453). We have found that after 3 weeks of oral mannan immunization no significant differences in ASCA levels were obtained in the serum comparing PBS versus mannan-treated mice (Supplementary Fig. 8b). Accordingly, after DSS-induced colitis, no major alterations were observed in the disease activity index, or in colon length comparing mannan-oral treated versus non-treated mice (Supplementary Fig. 8c,d). This result suggests that oral supplementation with mannan (at least in our experimental setting) is not sufficient to induce higher titers of serum ASCA associated with colitis susceptibility. This may be explained by the fact that stomach barrier (pH, motility, etc) may impose some roadblocks in the capacity of mannan to reach the intestinal mucosa, being thus not effective in changing mucosa immune response. Increased amounts of mannan or a different scheme of administration may be needed to further explore, in future studies, the impact of oral mannan exposure in ASCA production and possible impact in colitis susceptibility.

We have added these results to the manuscript in a novel figure, Supplementary Figure 8, as well as in the Results section (line 425) and in the methodology (line 810):

Results:

To further assess whether oral supplementation with mannan would be able to promote similar susceptibility to colitis, we have treated mice with mannan by oral gavage, followed by DSS-induced colitis (Supplementary Fig. 8a). Oral immunization with mannan was not effective in inducing increased serum ASCA IgG levels in comparison with non-treated mice (Supplementary Fig. 8b) and, thus expectedly, no major alterations were found on colitis susceptibility (Supplementary Fig. 8c) nor in colon length (Supplementary Fig. 8d), which clearly demonstrates that increased levels of ASCA IgGs are intricately related to increase susceptibility to colitis.

Methods:

To assess the effect of oral mannan treatment, an immunization protocol was performed during 3 weeks, in which mice were treated with 12 mg of mannan by oral gavage in the first 3 days of each week (total of 9 administrations), followed by DSS-induced colitis.

Version 1:

Decision Letter:

Our ref: NI-A36100A

1st May 2024

Dear Dr. Pinho,

Thank you for submitting your revised manuscript "A unique serum glycosylation signature of IgGs predicts the development of Crohn's disease and is associated with pathogenic anti-mannose glycan (ASCA) antibodies" (NI-A36100A). It has now been seen by the original referees and their comments are below. The reviewers find that the paper has improved in revision, and therefore we'll be happy in principle to publish it in Nature Immunology, pending minor revisions to satisfy the referees' final requests and to comply with our editorial and formatting guidelines.

We will now perform detailed checks on your paper and will send you a checklist detailing our editorial and formatting requirements in about a week. Please do not upload the final materials and make any revisions until you receive this additional information from us.

If you had not uploaded a Word file for the current version of the manuscript, we will need one before beginning the editing process; please email that to immunology@us.nature.com at your earliest convenience.

Thank you again for your interest in Nature Immunology Please do not hesitate to contact me if you have any questions.

Sincerely,

Stephanie Houston, PhD
Senior Editor
Nature Immunology

Reviewer #1 (Remarks to the Author):

The authors have addressed as far as feasible in this complicated setup my concerns. I have no further comments.

Reviewer #2 (Remarks to the Author):

The authors sufficiently addressed all of my comments and I wish to congratulate them on a beautiful piece of important work.

Well done!

Version 2:

Decision Letter:

In reply please quote: NI-A36100B

Dear Dr. Pinho,

I am delighted to accept your manuscript entitled "A unique serum glycosylation signature of IgGs predicts the development of Crohn's disease and is associated with pathogenic anti-mannose glycan (ASCA) antibodies" for publication in an upcoming issue of *Nature Immunology*.

Over the next few weeks, your paper will be copyedited to ensure that it conforms to *Nature Immunology* style. Once your paper is typeset, you will receive an email with a link to choose the appropriate publishing options for your paper and our Author Services team will be in touch regarding any additional information that may be required.

Please note that *Nature Immunology* is a Transformative Journal (TJ). Authors may publish their research with us through the traditional subscription access route or make their paper immediately open access through payment of an article-processing charge (APC). Authors will not be required to make a final decision about access to their article until it has been accepted. [Find out more about Transformative Journals](https://www.springernature.com/gp/open-research/transformative-journals).

Your paper will be published online soon after we receive your corrections and will appear in print in the next available issue.

Also, if you have any spectacular or outstanding figures or graphics associated with your manuscript - though not necessarily included with your submission - we'd be delighted to consider them as candidates for our cover. Simply send an electronic version (accompanied by a hard copy) to us with a possible cover caption enclosed.

You can now use a single sign-on for all your accounts, view the status of all your manuscript submissions and reviews, access usage statistics for your published articles and download a record of your refereeing activity for the *Nature* journals.

If you have not already done so, we strongly recommend that you upload the step-by-step protocols used in this manuscript to protocols.io. protocols.io is an open online resource that allows researchers to share their detailed experimental know-how. All uploaded protocols are made freely available and are assigned DOIs for ease of citation. Protocols can be linked to any publications in which they are used and will be linked to from your article. You can also establish a dedicated workspace to collect all your lab Protocols. By uploading your Protocols to protocols.io, you are enabling researchers to more readily reproduce or adapt the methodology you use, as well as increasing the visibility of your protocols and papers. Upload your Protocols at <https://protocols.io>. Further information can be found at <https://www.protocols.io/help/publish-articles>.

Please note that we encourage the authors to self-archive their manuscript (the accepted version before copy editing) in their institutional repository, and in their funders' archives, six months after publication. Nature Portfolio recognizes the efforts of funding bodies to increase access of the research they fund, and strongly encourages authors to participate in such efforts. For information about our editorial policy, including license agreement and author copyright, please visit www.nature.com/ni/about/ed_policies/index.html

Sincerely,

Stephanie Houston, PhD
Senior Editor
Nature Immunology

Click here if you would like to recommend Nature Immunology to your librarian
<http://www.nature.com/subscriptions/recommend.html#forms>

** Visit the Springer Nature Editorial and Publishing website at http://editorial-jobs.springernature.com?utm_source=ejp_NImm_email&utm_medium=ejp_NImm_email&utm_campaign=ejp_NImm for more information about our career opportunities. If you have any questions please click [here](mailto:editorial.publishing.jobs@springernature.com).**
